# Interpretability Is in the Mind of the Beholder: A Causal Framework for Human-Interpretable Representation Learning

**DOI:** 10.3390/e25121574

**Published:** 2023-11-22

**Authors:** Emanuele Marconato, Andrea Passerini, Stefano Teso

**Affiliations:** 1Dipartimento di Ingegneria e Scienza dell’Informazione, University of Trento, 38123 Trento, Italy; emanuele.marconato@unitn.it (E.M.); andrea.passerini@unitn.it (A.P.); 2Dipartimento di Informatica, University of Pisa, 56126 Pisa, Italy; 3Centro Interdipartimentale Mente/Cervello, University of Trento, 38123 Trento, Italy

**Keywords:** explainable AI, causal representation learning, alignment, disentanglement, causal abstractions, concept leakage

## Abstract

Research on Explainable Artificial Intelligence has recently started exploring the idea of producing explanations that, rather than being expressed in terms of low-level features, are encoded in terms of *interpretable concepts learned from data*. How to reliably acquire such concepts is, however, still fundamentally unclear. An agreed-upon notion of concept interpretability is missing, with the result that concepts used by both post hoc explainers and *concept-based* neural networks are acquired through a variety of mutually incompatible strategies. Critically, most of these neglect the human side of the problem: *a representation is understandable only insofar as it can be understood by the human at the receiving end*. The key challenge in human-interpretable representation learning (hrl) is how to model and operationalize this human element. In this work, we propose a mathematical framework for acquiring *interpretable representations* suitable for both post hoc explainers and concept-based neural networks. Our formalization of hrl builds on recent advances in causal representation learning and explicitly models a human stakeholder as an external observer. This allows us derive a principled notion of *alignment* between the machine’s representation and the vocabulary of concepts understood by the human. In doing so, we link alignment and interpretability through a simple and intuitive *name transfer* game, and clarify the relationship between alignment and a well-known property of representations, namely *disentanglement*. We also show that alignment is linked to the issue of undesirable correlations among concepts, also known as *concept leakage*, and to content-style separation, all through a general information-theoretic reformulation of these properties. Our conceptualization aims to bridge the gap between the human and algorithmic sides of interpretability and establish a stepping stone for new research on human-interpretable representations.

## 1. Introduction

The field of Explainable Artificial Intelligence (XAI) has developed a wealth of attribution techniques for unearthing the reasons behind the decisions of black-box machine learning models [1]. Traditionally, explaining a prediction involves identifying and presenting those low-level *atomic elements*—like input variables [2,3] and training examples [4,5]—that are responsible for said prediction (in the following, we will use the terms “responsibility” and “relevance” interchangeably). Explanations output by white-box models, such as sparse linear classifiers [6] and rule-based predictors [7], follow the same general setup. These atomic elements, however, are not very expressive and, as such, can be ambiguous [8]. To see this, consider an image of a red sports car that is tagged as “positive” by a black-box predictor. In this example, a saliency map would highlight those *pixels* that are most responsible for this prediction: these do not say whether the prediction depends on the image containing a “car”, on the car being “red”, or on the car being “sporty”. As a consequence, it is impossible to understand what the model is “thinking” and how it would behave on other images based on this explanation alone [9].

This is why focus in XAI has recently shifted toward explanations expressed in terms of higher-level symbolic representations, or *concepts* for short. These promise to ensure explanations are rich enough they can capture the machine’s reasoning patterns, while being expressed in terms that can be naturally understood by stakeholders [8,10].

This trend initially emerged with (post hoc) *concept-based explainers* (CBEs) like TCAV [11] and Net2Vec [12], among others [13,14,15], which match the latent space of a deep neural network to a vocabulary of pre-trained concept detectors. The idea of using higher-level concepts was foreshadowed in the original LIME paper [3]. These were quickly followed by a variety of *concept-based models* (CBMs)—including Self-Explainable Neural Networks [16], Part-Prototype Networks [17], Concept-Bottleneck Models [18], GlanceNets [19], and Concept Embedding Models [20]—that support representation learning while retaining interpretability. Specifically, these approaches learn a neural mapping from inputs to concepts, and then leverage the latter for both computing predictions—in a simulatable manner [21]—and providing ante-hoc explanations thereof. See [22] for a review. Since concepts act as a *bottleneck* through which all information necessary for inference must flow, CBMs hold the promise of avoiding the lack of faithfulness typical of post hoc techniques, while enabling a number of useful operations such as interventions [18] and debugging [23,24] using concepts as a human-friendly interface.

### 1.1. Limitations of Existing Works

The premise of conceptual explanations rests on the assumption that learned concepts are themselves interpretable. This begs the question: *what does it mean for a vocabulary of concepts to be interpretable?*

Researchers have proposed a variety of practical strategies to encourage the interpretability of the learned concepts, but no consistent recipe. Some CBMs constrain their representations according to intuitive heuristics, such as similarity to concrete training examples [17] or activation sparsity [16]. However, the relationship between these properties and interpretability is unclear, and unsurprisingly, there are well-known cases in which CBMs acquire concepts activating on parts of the input with no obvious semantics [25,26]. A more direct way of controlling the semantics of learned concepts is to leverage *supervision* on the concepts themselves, a strategy employed by both CBEs [11] and CBMs [18,19,27]. Unfortunately, this is no panacea, as doing so cannot prevent *concept leakage* [28,29], whereby information from a concept “leaks” into another, seemingly unrelated concept, compromising its meaning.

At the same time, concept quality is either assessed qualitatively in a rather unsystematic fashion—e.g., by inspecting the concept activations or saliency maps on a handful of examples—or quantitatively, most often by measuring how well-learned concepts match annotations. This so-called *concept accuracy*, however, is insufficient to capture issues like concept leakage.

Besides these complications, existing approaches neglect a critical aspect of this learning problem: that *interpretability is inherently subjective*. For instance, explaining a prediction to a medical doctor requires different concepts than explaining it to a patient: the notion of “intraepithelial” may be essential for the former, while being complete gibberish to the latter. However, even when concept annotations are employed, they are gathered from offline repositories and as such they may not capture concepts that are meaningful to a particular expert, or that despite being associated with a familiar name, follow semantics incompatible with those the user attaches to that name. Of course, there are exceptions to this rule. These are discussed in Section 6.

### 1.2. Our Contributions

Motivated by these observations, *we propose to view interpretability as the machine’s ability to communicate with a specific human-in-the-loop*. Specifically, we are concerned with the problem of learning conceptual representations that enable this kind of communication for both post and ante-hoc explanations. We call this problem **human-interpretable representation learning**, or hrl for short. Successful communication is essential for ensuring human stakeholders can understand *explanations* based on the learned concepts and, in turn, realizing the potential of CBEs and CBMs. This view is compatible with recent interpretations of the role of symbols in neuro-symbolic AI [10,30]. The key question is how to model this human element in a way that can be actually *operationalized*. We aim to fill this gap.

Our first contribution is a conceptual and mathematical model—resting on techniques from causal representation learning [31]—of hrl that *explicitly models the human-in-the-loop*.

As a second contribution, we leverage our formalization to develop an intuitive but sound notion of *alignment* between the conceptual representation used by the machine and that of the human observer. Alignment is strictly related to *disentanglement*, a property of learned representations frequently linked to interpretability [32,33], but also strictly *stronger*, in the sense that disentanglement alone is insufficient to ensure alignment. Later on, we will formally show that this follows from Proposition 1. We propose that alignment is key for evaluating interpretability of both CBEs and CBMs.

Our formalization improves on the work of Marconato et al. [19] and looks at three settings of increasing complexity and realism: (i) a simple but non-trivial setting in which the human’s concepts are *disentangled* (i.e., individual concepts can be changed independently from each other without interference). (ii) a more general setting in which the human’s concepts are constrained to be disentangled in blocks; and (iii) an unrestricted setting in which the human concepts can influence each other in arbitrary manners. In addition, we identify a and previously ignored link between interpretability of representations and the notion of *causal abstraction* [34,35,36].

As a third contribution, we formally show that *concept leakage* can be viewed as a lack of disentanglement, and therefore of alignment. This strengthens existing results and allows to reinterpret previous empirical observations [19,37].

As a fourth contribution, we discuss key questions arising from our mathematical framework, including whether perfect alignment is sufficient and necessary for interpretability, how to measure it, how to implement it in representation learning, and how to collect the necessary concept annotations.

### 1.3. Outline

The remainder of this paper is structured as follows. In the next section, we introduce prerequisite material, and then proceed in Section 3 to formalize the problem of human-interpretable representation learning and cast concept interpretability in terms of *alignment between representations*. Next, in Section 4, we analyze in depth the notion of alignment in three settings of increasing complexity and study its relationship to the issue of concept leakage, and then look at the consequences of our formalization in Section 5. Finally, we discuss related works in Section 6 and offer some concluding remarks in Section 7.

## 2. Preliminaries

In the following, we indicate scalar constants *x* in lower-case, random variables *X* in upper case, ordered sets of constants x and random variables X in bold typeface, and index sets I in calligraphic typeface. We also use the shorthand [n]:={1,…,n}. Letting X = $\left(X_{1},\ldots ,X_{n}\right)$ and I⊆[n], we write $X_{I}$$:=(X_{i}:i\in I)$ to indicate the ordered subset indexed by I and $X_{-I}:=X\setminus X_{I}$ to denote its complement, and abbreviate X∖$\left\{X_{i}\right\}$ as $X_{-i}$.

### 2.1. Structural Causal Models and Interventions

A *structural causal model* (SCM) is a formal description of the causal relationships existing between parts of a (stochastic) system [38,39]. Formally, an SCM C specifies a set of *structural assignments* encoding direct causal relationships between variables (as customary, we work with SCMs that are *acyclic*, *causally sufficient* (i.e., there are no external, hidden variables influencing the system), and *causally Markovian* (i.e., each variable $X_{i}$ is independent of its non-descendant given its parents in the SCM) [38]) in the form:(1)$$X_{i}\leftarrow f_{i}\left({\mathrm{Pa}}_{i},N_{i}\right)$$
where X = $\left(X_{1},\ldots ,X_{n}\right)$ are variables encoding the state of the system, ${\mathrm{Pa}}_{i}\subseteq X$ are the direct causes of $X_{i}$, and $N_{i}$ are noise terms. Variables without parents are *exogenous*, and play the role of inputs to the system, while the others are *endogenous*. The full state of the system can be sampled by propagating the values of the exogenous variables through the structural assignments in a top-down fashion. SCMs can be viewed as *graphs* in which nodes represent variables, arrows represent assignments, and noise variables are usually suppressed; see Figure 1.

Following common practice, we assume the noise terms to be mutually independent from each other and also independent from the variables not appearing in the corresponding structural equations; that is, it holds that $N_{i}⊥\;⊥N_{j}$ for all i≠j and $N_{i}⊥\;⊥X_{j}$ for all i,j. This is equivalent to assuming there are no hidden confounders. This assumption carries over to all SCMs used throughout the paper.

An SCM C describes both a *joint distribution*
p(X) = $\prod_{i}p\left(X_{i}|{\mathrm{Pa}}_{i}\right)$ and how this distribution *changes* upon performing *interventions* on the system. These are modifications to the system’s variables and connections performed by an external observer. Using Pearl’s do-operator [38], (atomic) interventions can be written as $do(X_{i}\leftarrow x_{i})$, meaning that the value of the variable $X_{i}$ is forcibly changed to the value $x_{i}$, regardless of the state of its parents and children. Carrying out an atomic intervention yields a *manipulated SCM* identical to C, except that all assignments to $X_{i}$ are deleted (i.e., the corresponding links in the graph disappear) and all occurrences of $X_{i}$ in the resulting SCM are replaced by the constant $x_{i}$. The resulting manipulated distribution is *p*(X$\;|\;do\left(X_{i}\leftarrow x_{i}\right))=1X_{i}=x_{i}\cdot \prod_{j\neq i}\;p\left(X_{j}|{\mathrm{Pa}}_{j}\right)$. Non-atomic interventions of the form $do(X_{I}\leftarrow x_{I})$ work similarly. Expectations of the form $E[\cdot |do\left(X_{j}\leftarrow x_{j}\right)]$ are just regular expectations evaluated with respect to the manipulated distribution.

### 2.2. Disentanglement

Central to our work is the notion of disentanglement [31,33,40] in both its two acceptations, namely *disentanglement of variables* and *disentanglement of representations*. We henceforth rely on the causal formalization given by Suter et al. [41] and Reddy et al. [42]. We refer the reader to those papers for more details.

Intuitively, *a set of variables G = $\left(G_{1},\ldots ,G_{n}\right)$ is disentangled if the variables can be changed independently from one another*. For instance, if $G_{1}$ represents the “color” of an object and $G_{2}$ its “shape”, disentanglement of variables implies that changing the object’s color does not impact its shape. This should hold even if the variables G have a common set of parents C—playing the role of counfounders, such as sampling bias or choice of source domain [38]—meaning that they can be both disentangled *and* correlated (via C). From a causal perspective, disentanglement of variables can be defined as follows:

**Definition 1** (Disentanglement of variables).
*A set of variables G are disentangled if and only if p($G_{i}|$C, $do\left(G_{I}\leftarrow g_{I}^{\prime }\right))\equiv p(G_{i}\;|\;$C) for all possible choices of I⊆[n]∖{i} and $g_{I}^{\prime }$.*


Now, consider the SCM in Figure 1 (left). It is easy to see that the variables G are disentangled: any intervention $do(G_{I}\leftarrow g_{I}^{\prime })$ breaks the links from C to $G_{I}$, meaning that changes to the latter will not affect $G_{i}$. In this case, the variables G are also conditionally independent from one another given C, or equivalently $G_{i}⊥\;⊥G_{j}|C$ for every i≠j.

Later on, we will be concerned with data generation processes similar to the one illustrated in Figure 1 (right). Here, a set of *generative factors* G = $\left(G_{1},\ldots ,G_{n}\right)$ with common parents C cause an observation X, and the latter is encoded into a *representation*
M = $\left(M_{1},\ldots ,M_{k}\right)$ by a machine learning model $p_{\theta }\left(M|X\right)$. Specifically, the explicit relation between M and G is obtained by marginalizing over the inputs X:(2)$$p_{\theta }\left(M|G\right):=E_{x∼p(X|G)}\left[p_{\theta }\left(M|x\right)\right]$$

This can also be viewed as a *stochastic map* α:g↦m. Maps of this kind are central to our discussion.

Since G is disentangled (see Definition 1), we can talk about *disentanglement of representations* for M. We say that *M is disentangled with respect to G* if, roughly speaking, each $M_{j}$ encodes information about at most one $G_{i}$ or, more precisely, *as long as $G_{i}$ is kept fixed, the value of $M_{j}$ does not change even when the remaining factors $G\setminus \{G_{i}\}$ are forcibly modified via interventions*. The degree by which a representation *violates* disentanglement of representations can be measured using the PIDA metric:

**Definition 2** (PIDA [41]). *Let $G_{i}$ be a generative factor and $M_{j}$ an element of the machine representation. PIDA measures how much fixing $G_{i}$ to a given value $g_{i}$ insulates $M_{j}$ from changes to the other generative factors $G_{-i}$, and it is defined as:*
(3)$$\mathrm{PIDA}\left(G_{i},M_{j}|g_{i},g_{-i}\right):=d\left(p_{\theta }\left(M_{j}|do\left(G_{i}\leftarrow g_{i}\right)\right),p_{\theta }\left(M_{j}|do\left(G_{i}\leftarrow g_{i},G_{-i}\leftarrow g_{-i}\right)\right)\right)$$*where d is a divergence. The original definition [41] fixes d to be the difference between means. Here, we slightly generalize PIDA to arbitrary divergences, as doing so can account for changes in higher-order moments too. The average worst case over all possible choices of $g_{i}$ and $g_{-i}$ is given by:*
(4)$$\mathrm{EMPIDA}\left(G_{i},M_{j}\right):=E_{g_{i}}\left[\max_{g_{-i}}\mathrm{PIDA}\left(G_{i},M_{j}|g_{i},g_{-i}\right)\right]$$

**Definition 3** (Disentanglement of representations).
*We say that a representation M is disentangled with respect to G if and only if $\max_{j}\min_{i}\mathrm{EMPIDA}\left(G_{i},M_{j}\right)$ is exactly zero.*


In other words, M is disentangled with respect to G if, for every $M_{j}$ there exists a $G_{i}$ such that fixing the latter *insulates*
$M_{j}$ from changes to the other generative factors $G_{-i}$. In Section 4, we will build on both types of disentanglement to derive our notion of alignment between representations.

Another important notion is that of *context-style separation*, which can be viewed as a special case of disentanglement of representations [43]. Let the generative factors G be partitioned into two disentangled sectors $G_{I}$ and $G_{-I}$, representing task-relevant information (content) and task-irrelevant factors of variations (style), respectively. Then, M satisfies content-style separation if the following holds:

**Definition 4** (Content-style separation).
*Let $\left(G_{I},G_{-I}\right)$ be two disentangled sectors. Then, M separates content from style if it can be partitioned into $\left(M_{J},M_{-J}\right)$ such that:*

(5)
$$\mathrm{EMPIDA}(G_{I},M_{J})=0$$



This means that, if the content $G_{I}$ is fixed, the machine representation $M_{J}$ is isolated from changes to the style $G_{-I}$. This property is asymmetrical: it holds even if $M_{-J}$
*is* affected by interventions to $G_{I}$. Also, there is no requirement that the elements of $M_{J}$ are disentangled with respect to $G_{I}$.

## 3. Human Interpretable Representation Learning

We are concerned with acquiring interpretable machine representations. Our key intuition is that a representation is only interpretable as long as it can be *understood by the human at the receiving end*. Based on this, we formally state our learning problem as follows:

**Definition 5** (Intuitive statement).
*Human-interpretable representation learning (*
hrl
*) is the problem of learning a (possibly stochastic) mapping between inputs $x\in R^{d}$ and a set of machine representations $z\in R^{k}$ that enables a machine and a specific human stakeholder to communicate using those representations.*


This mapping can be modeled without loss of generality as a conditional distribution $p_{\theta }\left(Z|X\right)$, whose parameters θ are estimated from data. While Definition 5 encompasses both CBEs and CBMs, the meaning of Z differs in the two cases, as we show next.

### 3.1. Machine Representations: The Ante-Hoc Case

CBMs are neural predictors that follow the generative process shown in Figure 2 (left). During inference, a CBM observes an input x, caused by generative factors G, and extracts a representation M, e.g., by performing *maximum a posteriori* inference [44], from the distribution $p_{\theta }\left(M|x\right)$ implemented as a neural network. In practice, the concept encoder can be implemented in various ways, e.g., as a multi-label convolutional neural network [27], as a variational auto-encoder [19], or even as a large language model [45]. This representation is partitioned into two subsets: *$M_{J}$ are constrained to be interpretable, while $M_{-J}$ are not*. As shown in Figure 2, only the interpretable subset is used for inferring a prediction $\hat{y}$, while $M_{-J}$, if present, is used for other tasks, such as reconstruction [19]. Specifically, the predicted concepts $M_{J}$ are fed to a simulatable top layer $p_{\theta }\left(Y|M\right)$, most often a sparse linear layer, from which an explanation can be easily derived. Assuming $M_{J}$ is in fact interpretable, CBMs can provide local *explanations*, summarizing what concepts are responsible for a particular prediction in an ante-hoc fashion and essentially for free [22,46]. For instance, if $p_{\theta }\left(Y|M_{J}\right)$ is a linear mapping with parameters $w_{yj}$, the explanation for predicting $\hat{y}$ is given by [16,17,18,19,27,47]:(6)$$E=\{\left(w_{\hat{y}j},m_{j}\right)\;:\;j\in J\}$$
where each concept activation $m_{j}$ is associated with a “level of responsibility” inferred from the top layer’s weights. Specific CBMs are outlined in Section 6.

Summarizing, in the case of CBMs, the concepts Z used for communicating with users (see Definition 5) are embodied by the interpretable machine representation $M_{J}$.

### 3.2. Machine Representations: The Post Hoc Case

For CBEs, the generative process is different; see Figure 2 (right). In this case, the internal representation M of the model mapping from inputs X to labels *Y is not required to be interpretable*. For instance, it might represent the state of the neurons in a specific layer. CBEs explain the reasoning process in a post hoc fashion by extracting the activations of *high-level concepts* $\hat{H}$ from M, and then inferring a concept-based explanation E specifying the contribution of each ${\hat{H}}_{i}$ to the model’s prediction, often in the same form as Equation (Equation 6).

Here, we are concerned with the interpretability of $\hat{H}$. Some approaches extract them by (indirectly) relying on concept annotations. For instance, TCAV [11] takes a set of linear classifiers, one for each concept, pre-trained on a densely annotated dataset, and then adapts them to work on machine representations M. Unsupervised approaches instead mine the concepts directly in the space of machine representations through a linear decomposition [13,14,15,48]. Specific examples are discussed in Section 6. In general, there is no guarantee that the symbolic and sub-symbolic representations $\hat{H}$ and M capture exactly the same information. This introduces a *faithfulness* issue, meaning that CBE explanations may not portray a reliable picture of the model’s inference process [11,48,49,50].

However, the issue we focus on is that the representation $Z=\hat{H}$ used by CBEs to communicate with users is, in fact, interpretable, regardless of whether it is also faithful.

### 3.3. From Symbolic Communication to Alignment

What makes symbolic communication possible? While a complete answer to this question is beyond the scope of this paper, we argue that communication becomes challenging unless the concepts Z with which the machine and the human communicate are “*aligned*”, in the sense that concepts having the same *name* share the same (or similar enough) *semantics*. Other factors contributing to interpretability, as well as some further remarks on whether alignment is sufficient and necessary, will be discussed in Section 5.

In order to formalize this intuition, we focus on the generative process shown in Figure 3. In short, we assume observations x (e.g., images or text observed during training and test) are obtained by mapping generative factors G$∼p^{*}\left(G|C\right)$ through a hidden ground-truth distribution $p^{*}\left(X|G\right)$.

The observations x are then received by *two observers*: a machine and a human. The machine maps them to its own learned representation M, which may or may not be interpretable. The interpretable representations Z, which correspond to $M_{J}$ for CBMs (see Section 3.1) and to $\hat{H}$ for CBEs (Section 3.2), are then derived from M.

At the same time, the human observer maps the same observations to its own vocabulary of concepts H. For instance, if x is an image portraying a simple object on a black background, h may encode the “color” or “shape” of that object, or any other properties deemed relevant by the human. The choice and semantics of these concepts depend on the background and expertise of the human observer and possibly on the downstream task the human may be concerned with (e.g., medical diagnosis or loan approval) and, as such, may vary between subjects. It is *these* concepts that the human associates names, like in Figure 4, and it is these concepts that they would use for communicating the properties of x to other people.

Notice that the human concepts H may be arbitrarily different from the ground-truth factors G: whereas the latter include all information necessary to determine the observations, and as such may be complex and uninterpretable [51], the former are those aspects of the observation that matter *to the human observer*. A concrete example is that of color blindness: an observer may be unable to discriminate between certain wavelengths of visible light, despite these being causes of the generated image X. Another, more abstract, example is the generative factors that cause a particular apple to appear ripe, e.g., those biological processes occurring during the apple tree’s reproductive cycle, which are beyond the understanding of most non-experts. They are so opaque that a whole science had to be developed to identify and describe them. In stark contrast, the concept of “redness” is not causally related to the apple’s appearance, and yet easily understood by most human observers, precisely because it is a feature that is evolutionarily and culturally useful to those observers. In this sense, *the concepts H are understandable, by definition, to the human they belong to*.

We argue that symbolic communication is feasible whenever the names associated (by the human) to elements of H can be transferred to the elements of Z in a way that preserves semantics. That is, *concepts with the same name should have the same meaning*. In order to ensure information expressed in terms of Z—say, an explanation stating that $Z_{1}$ is irrelevant for a certain prediction—is understood by the human observer, we need to make sure that Z itself is somehow “aligned” with the human’s representation H.

## 4. Alignment as Name Transfer

### 4.1. Alignment: The Disentangled Case

What does it mean for two representations to be aligned? We start by looking at the simplest (but non-trivial) case in which the ground-truth factors G are *disentangled*; see Definition 1. For ease of exposition, let us also temporarily assume that some of the generative factors are inherently interpretable, as in [19]. Namely, we assume all factors in $G_{I}\subseteq G$, where I⊆[n], can be understood by the human observer, while those in $G_{-I}$ cannot. The corresponding data generation process is illustrated in Figure 4. Under these assumptions, we aim to recover machine representations M that are aligned to the interpretable factors $G_{I}$.

To this end, we generalize the notion of alignment introduced by Marconato et al. [19]. Our definition extends that of [19] to the general case in which the mapping α, which is defined as a marginal distribution in Equation (Equation 2), is stochastic rather than deterministic. Doing so allows us to cater to more realistic applications and to draw an explicit connection with PIDA in Proposition 1. As anticipated, our definition revolves around the conditional distribution on M given by G or, equivalently, the stochastic map α:g↦m defined in Equation (Equation 2) and shown in red in Figure 4. The key intuition is that *two concept vocabularies G and M are aligned if and only if α preserves the semantics of the interpretable generative factors $G_{I}$*.

More specifically, alignments holds if α allows to *transfer the names* of the interpretable factors in a way that preserves semantics. If $p_{\theta }\left(M|X\right)$ is learned in an unsupervised fashion, names are generally transferred by collecting or constructing inputs annotated with the corresponding human concepts, feeding them to the concept extractor, and looking for matches between the annotations and the elements of $M_{J}$. If concept-level annotations are used, the names are automatically transferred along with them, but we still wish the user to be able to match the learned concepts with its own. In a sense, this process is analogous to giving the human observer access to a set of “knobs”, each one controlling the value of one $G_{i}\in G_{J}$, and to a visualization of the machine representation $M_{I}$. Turning a knob is akin to *intervening* on the corresponding factor $G_{i}$. If, by turning a knob, the user is able to figure out what $G_{i}$ corresponds to what $M_{j}$, then they will associate them with the same name. Since we are assuming $G_{I}$ is disentangled, turning one knob does not affect the others, which simplifies the process.

The formal definition of alignment is as follows:

**Definition 6** (Alignment).
*Given generative factors G of which $G_{I}$ are interpretable, a machine representation M is aligned if the map α between G and M can be written as:*

(7)
$$M_{J}=\alpha {\left(G,N\right)}_{J}=\left(\mu_{j}\left(G_{\pi (j)},N_{j}\right)\;:\;j\in J\right)$$

*where $M_{J}\subseteq M$ are the machine representations that ought to be interpretable, N are independent noise variables, and π and μ satisfy the following properties:*
***D1***.
*The index map π:J↦I is surjective and, for all j∈J, it holds that, as long as $G_{\pi (j)}$ is kept fixed, $M_{j}$ remains unchanged even when the other generative factors $G\setminus \{G_{\pi (j)}\}$ are forcibly modified.*
***D2***.
*Each element-wise transformation $\mu_{j}$, for j∈J, is monotonic in expectation over $N_{j}$:*

(8)
$$\exists ⋈\in \left\{>,<\right\}\;s.\;t.\;\forall g_{\pi (j)}^{\prime }>g_{\pi (j)},\;\left(E_{N_{j}}\left[\mu_{j}\left(g_{\pi (j)},N_{j}\right)\right]-E_{N_{j}}\left[\mu_{j}\left(g_{\pi (j)}^{\prime },N_{j}\right)\right]\right)⋈0$$




Let us motivate our two desiderata. In line with prior work on disentangled representations [32,33], **D1** requires that α should not “mix” multiple $G_{i}$’s into a single $M_{j}$, regardless of whether the former belong to $G_{I}$ or not. For instance, if $M_{j}$ blends together information about both color and shape, or about color and some uninterpretable factor, human observers would have trouble pinning down which one of their concepts it matches. If it does not, then turning the $G_{\pi (j)}$ knob only affects $M_{j}$, facilitating name transfer. The converse is not true: as we will see in Section 4.4, interpretable concepts with “compatible semantics” can in principle be blended together without compromising interpretability. We will show in Section 4.2 that this is equivalent to disentanglement.

**D2** is also related to name transfer. Specifically, it aims to ensure that, whenever the user turns a knob $G_{\pi (j)}$, they can easily understand *what* happens to $M_{j}$ and thus figure out the two variables encode the same information. To build intuition, notice that both **D1** and **D2** hold for the *identity* function, as well as for those maps α that *reorder* or *rescale* the elements of $G_{I}$, which clearly preserve semantics and naturally support name transfer. Monotonicity captures all of these cases and also more expressive *non-linear* element-wise functions, while *conservatively* guaranteeing a human would be able to perform name transfer. Notice that the mapping needs not to be exact, in the sense that the output can depend on independent noise factors N. This leaves room for stochasticity due to, e.g., variance in the concept learning step. Notice also that **D2** can be constrained further based on the application.

A couple of remarks are in order. First, of all, while we have defined alignment between interpretable factors $G_{I}$ and the machine representation $M_{J}$, the same definition works even if we replace the former with the user’s concepts H and the latter with the concepts $\hat{H}$ extracted by a concept-based explainer. In both cases, α is the map between human concepts h and either $m_{J}$ or $\hat{h}$, and it is obtained by marginalizing over X, G, and C (see Figure 3). More generally, alignment can hold for *any* mapping between representations. We also observe that, since π maps J exclusively into I, alignment entails a form of *content-style separation* (Definition 4), in that $M_{J}$ does not encode any information about $G_{-I}$. We will show in Section 4.3 that representations that do not satisfy this condition can be affected by *concept leakage*, while aligned representations cannot. Finally, we note that M can be aligned and still contain multiple transformations of the same $G_{i}\in G_{I}$. This does not compromise interpretability in that all “copies” can always be traced back to the same $G_{i}$. Practical considerations on how to measure *alignment* are discussed in Section 5.2.

### 4.2. Disentanglement Does Not Entail Alignment

Next, we clarify the relationship between alignment and disentanglement of representations by showing that the latter is exactly equivalent to **D1**:

**Proposition 1.** 
*Assuming noise terms are independent, as per Section 2,*
*
**D1**
*
*holds if and only if the representations are disentangled in $\left(G_{I},M_{J}\right)$ (see Definition 3).*


All proofs can be found in Appendix A. The equivalence between disentanglement of representations and **D1** implies that *disentanglement is insufficient for interpretability*: even if M is disentangled, i.e., each $M_{j}$ encodes information about at most one $G_{i}\in G_{I}$, nothing prevents the transformation from $G_{i}$ to its associated $M_{j}$ from being arbitrarily complex, complicating name transfer. In the most extreme case, $\alpha {\left(\cdot \right)}_{j}$ may not be *injective*, making it impossible to distinguish between different $g_{i}$s, or could be an arbitrary shuffling of the continuous line: this would clearly obfuscate any information present about $G_{i}$. This means that, during name transfer, a user would be unable to determine what value of $M_{j}$ corresponds to what value of $G_{i}$ or to anticipate how changes to the latter affect the former.

This is why **D2** in Definition 6 requires the map between each $G_{i}\in G_{I}$ and its associated $M_{j}$ to be “simple”. This extra desideratum makes alignment *strictly stronger* than disentanglement.

### 4.3. Alignment Entails No Concept Leakage

*Concept leakage* is a recently discovered phenomenon whereby the “interpretable” concepts $M_{J}$ unintentionally end up encoding information about extraneous concepts [29]. Empirically, leaky concepts are predictive for inference tasks that, in principle, do not depend on them. Situations like the following occur in practice, even if full concept supervision is used [19,28,29]:

**Example 1.** *Let X be a* dSprites *image [52] picturing a white sprite, determined by generative factors including “*position*”, “*shape*”, and “*size *”, on a black background. Now imagine training a concept extractor $p_{\theta }\left(M|X\right)$ so that $M_{J}$ encodes* shape *and* size *(but not* position*) by using full concept-level annotations for* shape *and* size*. The concept extractor is then frozen. During inference, the goal is to classify sprites as either positive (Y=1) or negative (Y=0) depending on whether they are closer to the top-right corner or the bottom-left corner. When concept leakage occurs, the label, which clearly depends only on* position*, can be predicted with above random accuracy from $M_{J}$, meaning these concepts somehow encode information about* position*, which they are not supposed to.*

The issue with concept leakage is that it muddles the semantics of concepts $M_{J}$, which contain information they are not supposed to encode, and therefore of explanations built on them. Imagine that the learned concept for “red” also activates on a few objects that are, in fact, blue, due to leakage. Also, assume the predictor predicts a blue object as positive because red fires. Then, an explanation for that prediction would be that the blue object is positive because (according to the model) it is red. Clearly, this hinders trustworthiness. The only existing formal account of concept leakage was provided by Marconato et al. [19], who view it in terms of (lack of) out-of-distribution (OOD) generalization. Other works instead focus on in-distribution behavior and argue that concept leakage is due to encoding discrete generative factors using a continuous representation [37,53]. We go beyond these works by providing the first general formulation of concept leakage and showing that it is related to alignment. Specifically, we propose to view *concept leakage as a (lack of) content-style separation*, and show that this explains how concept leakage can arise both in- *and* out-of-distribution. In order to do so, we start by proposing a general reformulation of the concept leakage problem (Definition 7) and derive two bounds from mutual information properties (Proposition 2). Then, we show that a model that achieves perfect alignment avoids concept leakage entirely (Proposition 3 and Corollary 1).

We start by formalizing the intuition that concept leakage is excess prediction accuracy—gained by leveraging leaky concepts—compared to a leak-free baseline [19,53]. The corresponding generative process is reported in Figure 5. We assume the generative factors G are partitioned as $\left(G_{I},G_{-I}\right)$ such that *only*
$G_{-I}$ are *informative* for predicting a label *Y*, mediated by the conditional distribution $p(Y|G_{-I})$. This implies that their mutual information is positive; that is, $I(G_{-I},Y)>0$. In this section, we are mostly concerned with the non-informativeness of $G_{I}$, hence we allow $G_{-I}$ to potentially contain also interpretable factors. Now, fix a concept encoder $p_{\theta }\left(M_{J}|X\right)$ and let $q_{\lambda }\left(Y|M_{J}\right)$ be a predictor learned on top of it (in orange in the figure). *To quantify concept leakage, we look at how well the best possible such predictor can infer the label Y using $M_{J}$* after intervening on $G_{-I}$. Analogously to EMPIDA (Definition 2), the intervention detaches $G_{-I}$ from C, thus ensuring the label *Y* cannot be influenced by the irrelevant factors $G_{I}$. The resulting manipulated distribution on G is:(9)$$p^{\prime }\left(G\right)=p\left(G|do\left(G_{-I}\leftarrow g_{-I}\right)\right)q\left(g_{-I}\right):=E_{C}\left[p\left(G_{I}|C\right)\right]1G_{-I}=g_{-I}q\left(g_{-I}\right)$$
where $q(g_{-I})$ is a distribution over possible interventions. This can be *any* distribution, with the only requirement that under any intervention $do(G_{-I}\leftarrow g_{-I})$, the model observes different variations in *Y*. If *Y* is constant, leakage is impossible, since $I(G_{-I},Y)=0$.

From the causal factorization in Figure 5, the joint probability of (X,Y) resulting from the post-interventional distribution $p^{\prime }\left(G\right)$ is given by: (10)$$p\left(X,Y\right)=E_{g∼p^{\prime }(G|do([G}-I\leftarrow g_{-I}))$$

Data of this kind appear, for example, in the dSprites experiment [19] outlined in Example 1. Here, during training, the “position” of the sprite is fixed (i.e., $G_{pos}=G_{-I}$ are fixed to the center), while at test time the data contains different interventions over the position $G_{pos}=G_{-I}$, and free variations in the other factors $G_{I}$ (e.g., “shape” and “size”). Essentially, these interventions move the sprite around the top-right and bottom-left borders, where the factors $G_{pos}$ are extremely informative for the label *Y*.

In order to measure the degree of concept leakage in $p_{\theta }\left(M_{J}|X\right)$,
*
we compare the prediction performance of the best possible predictor $q_{\lambda }\left(Y|M_{J}\right)$ with that of the best possible predictor $r_{\gamma }\left(Y\right)$ that does not depend on $M_{J}$ at all*. This is equivalent to comparing the behavior of two Bayes optimal predictors, one of which has access to the learned (possibly leaky) concepts whereas the other does not. In the following, we assume the distributions $q_{\lambda }$ an $r_{\gamma }$ to be sufficiently expressive, i.e., they can encode any sufficiently well behaved stochastic function. This is the case, for instance, when they are implemented as deep neural networks. We are now ready to define concept leakage:

**Definition 7** (Concept leakage).*Given a classifier $q_{\lambda }\left(y|z\right)$, an uninformed Bayes optimal predictor $r_{\gamma }\left(y\right)$, and data samples (x,y)∈D, concept leakage Λ is the difference:*(11)$$\Lambda =\max_{\lambda }\left[L_{CL}\left(\lambda \right)\right]-\max_{\gamma }\left[L_{r}\left(\gamma \right)\right]$$*where:*(12)$$L_{CL}=E_{\left(x,y)∼p(X,Y\right)}\log q_{\lambda ,\theta }\left(y|x\right)\;L_{r}=E_{\left(x,y)∼p(X,Y\right)}\log r_{\gamma }\left(y\right)$$*are the average log-likelihood of the classifier*$q_{\lambda ,\theta }\left(Y|X\right):=E_{m_{J}∼p_{\theta }\left(M_{J}|X\right)}p\left(Y|m_{J}\right)$*and of the uninformed Bayes optimal classifier, respectively*.

By Definition 7, concept leakage occurs if and only if there exists a λ that allows to predict *Y* better than the best uninformed predictor. In the following analysis, we characterize concept leakage evaluated on the ground-truth distribution p(X,Y). We proceed to show that this quantity is bounded by two terms:

**Proposition 2.** 
*Assuming the causal factorization in Figure 5, it holds that:
*

(13)
$$I\left(M_{J},Y\right)\leq \Lambda \leq I\left(G_{-I},Y\right)$$

*where I(A,B) denotes the mutual information between A and B.
*


The bounds in Equation (Equation 13) are useful for understanding how concept leakage behaves. They show, for instance, that Λ cannot exceed the mutual information between $G_{-I}$ and *Y*. Second, applying the data-processing inequality [54] to the lower bound yields $I\left(M_{J},Y\right)\geq I\left(M_{J},G_{-I}\right)$. The latter quantifies the information contained in $M_{J}$ about $G_{-I}$. In other words, concept leakage can only be zero if indeed the machine concepts $M_{J}$ contain no information about them, because $I(M_{J},G_{-I})$≤Λ=0. Next, we also show that if $M_{J}$ does not encode information about $G_{-I}$—or equivalently, it satisfies content-style separation (Definition 4)—then it has zero concept leakage.

**Proposition 3.** 
*
Suppose that $M_{J}$ does not encode any information of $G_{-I}$, consistently with content-style separation (Definition 4), then Λ is zero.
*


This result leads to two consequences. Let us start by looking at the
*
out-of-distribution case
*
investigated in [19]. Here, the concept extractor is trained only on some fixed variations of $G_{-I}$. However, when the support of $G_{-I}$ changes drastically, the model is not likely to ensure content-style separation outside of the support of the training distribution, even if D1 holds in-distribution. This failure can be explained by the difficulty of disentanglement techniques to ensure disentanglement for out-of-distribution samples, in the context of combinatorial generalization, by Montero et al. [55], Montero et al. [56]. Consider the dSprites example. Here, during training, sprites are located in the dead center of the background, and when observing the sprites on the borders of the image, which is far away from the support of the training set, the concept encoder fails to ensure their representations are disentangled. This failure of disentanglement techniques to ensure disentanglement for out-of-distribution inputs was also observed, in the context of combinatorial generalization, by Montero et al. [55], Montero et al. [56]. Our results show that if content-style separation does not hold, concept leakage may be non-zero, meaning that techniques like open-set recognition [57] must be adopted to detect OOD inputs and process them separately.

Next, we look at concept leakage for *in-distribution* scenarios. Following Havasi et al. [53], consider a model leveraging two concepts—the presence of “tail” and “fur”—and the task of distinguish between images of cats and dogs using these (clearly non-discriminative) concepts. According to [53], concept leakage can occur when binary concepts like these are modeled using continuous variables, meaning the concept extractor can unintentionally encode “spurious” discriminative information. In light of our analysis, we argue that concept leakage is instead due to lack of content-style separation, and thus of alignment. To see this, suppose there exists a concept $G_{k}\in G_{-I}$ useful for distinguishing cats from dogs and that it is disentangled as in Definition 1 from the concepts of fur $G_{fur}$ and of tail $G_{tail}$. Then, by content-style separation, any representation $M_{J}$ that is aligned to $G_{fur}$ and $G_{tail}$ does not encode any information about $G_{k}$, leading to zero concept leakage.

In both cases, concept leakage arises as a failure in content-style separation between relevant and irrelevant generative factors, and as such it can be used as a proxy for measuring the latter. Moreover, since alignment implies content-style separation, aligned representations cannot suffer from concept leakage. Note that the converse is not true: while alignment entails content-style separation, the latter can hold independently from alignment.

In-distribution concept leakage can also be extended to encompass the case where concepts belong to interpretable concepts $G_{I}$. This is the original context considered by Havasi et al. [53] where, for instance, $\left(G_{fur},G_{tail},G_{k}\right)\subseteq G_{I}$. Again, we suppose that only the ground-truth factor $G_{k}$ is relevant for in-distribution predictions *Y* (being a cat or a dog). In this case, concept leakage is evaluated among those elements of $M_{J}$ that should not encode $G_{k}$. Practically, if a subset of $M_{J}$ encodes only the concepts $G_{fur}$ and $G_{tail}$ it must not be discriminative for *Y*, otherwise complicating the semantics of the learned concepts. Without loss of generality (the general case includes all representations $M_{\pi^{-1}\left(k\right)}$, where $\pi^{-1}$ is the pre-image of the map π), we suppose that only a single $M_{j^{\prime }}$ is aligned to $G_{k}$, that is $\pi (j^{\prime })=k$, whereas other $M_{J}\setminus M_{j^{\prime }}$ are aligned to other concepts, among which $G_{fur}$ and $G_{tail}$. Then, the following holds:

**Corollary 1.** 
*Consider a representation $M_{J}$ that is aligned to a set of disentangled concepts $G_{I}$, among which only $G_{k}$ is discriminative for the label *Y*. Then, all $M_{j}\in M_{J}$ that are not associated by α to $G_{k}$, i.e., π(j)≠k, do not suffer from concept leakage.
*


Ultimately, an aligned representation prevents concept leakage within the encoded concepts $M_{J}$, guaranteed by having a *disentanglement* from **D1**. In fact, if the representations $\left(M_{j_{1}},M_{j_{2}}\right)$ are aligned to the concepts $G_{fur}$ and $G_{tail}$, respectively, they cannot be used to discriminate between *cats* and *dogs*.

### 4.4. Alignment: The Block-Wise Case

So far, we assumed the generative factors $G_{I}$ (or, equivalently, the human concepts H) are disentangled. We now extend alignment to more complex cases in which the human concepts
*
can
*
be mixed together without compromising interpretability. This covers situations in which, for instance, the machine captures a single categorical generative factor using multiple variables via one-hot encoding, or uses polar coordinates to represent the 2D position of an object.

To formalize this setting, we assume $G_{I}$ and $M_{J}$ are partitioned into non-overlapping “blocks” of variables $G_{I^{\prime }}\subseteq G_{I}$ and $M_{J^{\prime }}\subseteq M_{J}$, respectively. The idea is that each block $M_{J^{\prime }}$ captures information about only a single block $G_{I^{\prime }}$, and that while mixing across blocks is not allowed, mixing the variables within each block
*
is*. From the human’s perspective, this means that name transfer is now performed block by block. With this in mind, we define
*
block alignment
*
as follows:

**Definition 8** (Block-wise alignment).
*A machine representation M is block-wise aligned to $G_{I}$ if and only if there exists a subset $M_{J}\subseteq M$, a partition $P_{M}$ of J, and a mapping α:(g,N)↦m such that:*

(14)
$$M_{J^{\prime }}=\alpha {\left(G,N\right)}_{J^{\prime }}:=\mu_{J^{\prime }}\left(G_{\Pi (J^{\prime })},N_{J^{\prime }}\right)\;\forall J^{\prime }\in P_{M}$$

*
where the maps Π and μ satisfy the following properties.
*
***D1***

*
There exists a partition $P_{G}$ of I such that $\Pi :P_{M}\to P_{G}$. In principle, we can extend this notion to a family of subsets $P_{G}$ of I. As an example, for xyz positions, one can consider blocks {xy,yz,xz} that are mapped to, respectively, block aligned representations. We call this condition block-wise disentanglement.
*
***D2***

*
Each map $\mu_{J^{\prime }}$ is simulatable and invertible (for continuous variables, we require it to be a diffeomorphism) on the first statistical moment; that is, there exists a unique pre-image $\alpha^{-1}$ defined as:
*

(15)
$$G_{\Pi (J^{\prime })}=\alpha^{-1}{\left(E\left[M_{J}\right]\right)}_{J^{\prime }}:={\left(E_{N_{J}}\left[\mu_{J^{\prime }}\left(\cdot ,N_{J^{\prime }}\right)\right]\right)}^{-1}\left(G_{\Pi (J^{\prime })}\right)$$




By **D1**, changes to any block of human concepts only impact a single block of machine concepts, and by **D2** the change can be anticipated by the human observer, that is the human interacting with the machine grasps what is the general mechanism behind the transformation from G (or H) and M (and vice versa). Both properties support name transfer.

A priori, it is not clear to say what transformations are simulatable [21], as this property depends crucially on the human’s cognitive limitations and knowledge. We remark that **D2** intuitively constraints the variables within each block to be “semantically compatible”. In the context of image recognition, for instance, placing concepts such as “nose shape” and “sky color” in the same block is likely to make name transfer substantially more complicated, as changes to “nose shape” might end affecting the representation of “‘sky color” and vice versa. In that case, it would not be easy for a user to figure out how the concepts have been mixed, undermining simulatability. An example of semantic compatibility is that of rototranslation of the coordinates followed by element-wise rescaling. This condition is identical to “weak identifiability” in representation learning [58,59]. A counter example would be a map α given by a conformal map for the 2D position of an object in a scene. Albeit invertible, it may not be simple at all to simulate.

Notice that with this definition, we include two possible scenarios: (i) the case where some of the ground-truth concepts belonging to the same block are transformed in a single block; and (ii) the case where semantically compatible, but disentangled, concepts $G_{I}$ are mixed together in $M_{J}$, which is often neglected in current disentanglement literature. The latter includes and extends the special case of alignment for disentangled G.

The limitation of Definition 8 reflects the fact that taking into account the possible user’s grasp of the representation is not straightforward to define and poses a challenge to provide a uniquely accepted definition that considers the human factor.

### 4.5. Alignment: The General Case

In the most general case, the generative factors G are causally related to each other according to an arbitrary ground-truth SCM $C_{G}$. This entails that $G_{I}$ is no longer (block) disentangled. Hence, during name transfer, turning a “knob” (i.e., a variable $G_{i}$) affects those knobs that are causally dependent on it.

Naturally, the semantics of the user’s comprises the causal relations between them. To see this, let $G_{1}$ be the temperature and $G_{2}$ the color of a metal object: the user knows that temperature affects color, and would not assign the same name to a representation of temperature that does not have a similar effect on the representation of color. In order ensure preservation of these semantics, we say that a machine representation M is *aligned* to G if, whenever the human intervenes on one $G_{i}$, affecting those generative factors $G_{I^{\prime }}$ that depend on it, an analogous change occurs in the machine representation.

We now show that *block-alignment* is sufficient to satisfy this desideratum. Even in this more general case, the distribution over the representation can be obtained by marginalizing over the input variables:
(16)$$p\left(M\right):=E_{x∼p(X)}p_{\theta }\left(M|x\right)\equiv E_{g∼p(G)}p_{\theta }\left(M|g\right)$$
Notice that the definition of block alignment does not make any assumption about absence or presence of causal relations between blocks of generative factors, meaning that we it is still well-defined in this more general setting. This generalizes block alignment beyond disentangled factors [41]. Specifically, a map α can be block aligned if the variables *G* within each block are disentangled with each other, although there may exist causal relations across blocks.

Now, imagine having a stochastic map α between G and M that does satisfy block alignment, and also that there exist causal relations between the blocks $G_{I^{\prime }}$. Whenever the user turns a “knob” corresponding to a ground-truth block, this yields an interventional distribution $p(G|do\left(G_{I^{\prime }}\leftarrow g_{I^{\prime }}\right))$. Through α, this determines a new interventional distribution on the machine representations, namely:
(17)$$p\left(M|do\left(G_{I^{\prime }}\leftarrow g_{I^{\prime }}\right)\right)=E_{g∼p(G|do\left(G_{I^{\prime }}\leftarrow g_{I^{\prime }}\right))}p_{\theta }\left(M|g\right)$$

This implies a representation M where the (interventional) distribution is obtained by mapping the state $g_{I^{\prime }}$ through α. The same operation can be performed to obtain the state of all other machine representations aligned with the blocks that are causally related to $G_{I^{\prime }}$ and affected by the intervention.

Note that this distribution *automatically takes causal relations between generative factors into account and treats them as causal relations between machine representations*. To see this, consider the following example:

**Example 2.** *
Consider two generative factors $G_{1}$ and $G_{2}$ causally connected via a structural assignment $G_{2}\leftarrow f\left(G_{1},N_{2}\right)$, as in Figure 6. As before, $G_{1}$ could be the* temperature*and*$G_{2}$*the* color*of a metal solid. Correspondingly, the aligned representation* M *encodes the* temperature *in two distinct variables, $M_{1}$ and $M_{3}$, respectively, to the temperature, say, measured in degrees Celsius and degrees Fahrenheit. $M_{2}$ encodes the* color *variable*.*The consequence of block-wise alignment is sketched in Figure 6 in three distinct cases: (left) intervening on the temperature $G_{1}$ affects both the aligned variables $\left(M_{1},M_{3}\right)$ and the color $G_{2}$. Correspondingly, this also has an effect on $M_{2}$ that changes according to $G_{2}$. (center) An intervention on $G_{2}$ influences only $M_{2}$ through α and it does not affect $M_{1}$ and $M_{3}$. (right) The effect of an intervention on whole variables G is localized such that interventions on the temperature factor $G_{1}$ will affect $M_{1}$ and $M_{3}$, whereby the interventions on $G_{2}$ only affect $M_{2}$, isolating it from the intervention on $G_{1}$
*.

Next, we formalize our observation that, thanks to *block alignment*, interventions on G are automatically mirrored on M:

**Proposition 4.** *Given a block-wise aligned representation M to G, it holds that for each distinct block $M_{K}$ of representations M, an intervention on $G_{\Pi (K)}$ isolates $M_{K}$ from interventions of other ground-truth blocks $G_{-\Pi (K)}$. Moreover, distinct interventions on $G_{\Pi (K)}$ corresponds on average to different interventions on $M_{K}$
*.

Importantly, this means that the effect of an intervention on the whole G isolates each block in M from the others, i.e., there is no explicit causal relation appearing in the learned representation. This matches the intuition that an intervention on a specific generative factor affects the corresponding block and removes the dependencies on other blocks of the representation. Following Example 2, this means that the a intervention on the temperatureconcept will affect both the corresponded representations and the ones aligned to concept of color, whereas intervening on the latter would only amount to change the representation of color, irrespective of the value assumed in the temperature representation.

To summarize, block alignment entails interventions on the ground-truth concepts are mapped properly. At the same time, alignment between blocks ensures the transformation α is simulatable, meaning that users can understand changes happening to all of the variables involved. This is sufficient to guarantee name transfer can be completed successfully in the general case, assuming not too many factors are changed at a time.

### 4.6. Alignment and Causal Abstractions

One important observation is that the form of name transfer we have considered is
*
asymmetrical*, in the sense that the user intervenes on its own representation H only, to then check how this impacts M. The other direction is not considered: it is not necessary to consider how intervening on M impacts M. This leads to the setup depicted in Figure 7 (right) in which, given $C_{H}$, the effects of interventions on $H_{i}$ are propagated to M via a map β:H↦M, which may or may not be block-aligned.

We now consider a scenario in which the SCM of the representation $C_{M}$ is also provided (in practice, $C_{M}$ can be uncovered from data via causal discovery [38]) and the effects of interventions on M can be propagated leveraging its structural assignments.

Ideally, we would expect that, as long as M is block-aligned to H, we can always find analogous post-interventional effects when intervening on $H_{i}$ and on its aligned variable $M_{j}$. This underlies a consistency condition between the two “worlds” that are described with $C_{H}$ and $C_{M}$, respectively, by requiring that they both lead to similar conclusions when intervened in a equivalent manner. Clearly, this does not depend solely on the nature of the map β, but also on the structure of the machine SCM $C_{M}$.

The presence of a consistency property between $C_{H}$ and $C_{M}$ is what defines a
*
causal abstraction
*
[34,35,60]; see [61] for an overview. Causal abstractions have been proposed to define (approximate) equivalence between causal graphs and have recently been employed in the context of explainable AI [36,62]. The existence of a causal abstraction ensures two systems are
*
interventionally equivariant*: interventions on one system can always be mapped (modulo approximations) to equivalent interventions in the other and lead to the same interventional distribution.

All causal abstractions check the consistency between two maps under the same intervention $do(H_{I})$: one is defined by the post-interventional distribution of $C_{H}$ that is mapped on M via β, the other one consists of first matching on M the correspondent action $do(M_{J})$ and propagating it via $C_{M}$. Intuitively, this means that, under β, interventions on H lead to the same conclusion as interventions on M. We formalize this idea from constructive causal abstractions (existing works on causal abstractions [34,62] do not impose the map between values or interventions are simulatable, meaning that even if $C_{M}$ is a causal abstraction of $C_{H}$, it may be impossible for users to understand the mapping between the two) [62] by adapting it to the case where H and M are connected by block-alignment:

**Definition 9** (β-aligned causal abstraction).
*The $C_{M}$ is a causal abstraction of $C_{H}$ under block-alignment β if, for all possible interventions $do(H_{I}\leftarrow h_{I})$ with $H_{I}\subseteq H$, the following diagram commutes:*

(18)
$$\begin{array}{ccc} do\left(H_{I}\leftarrow h_{I}\right) & \overset{C_{H}}{\to } & p\left(H|do(H_{I}\leftarrow h_{I})\right)\\ ↓\beta & & ↓\beta_{*}\\ do\left(M_{J}\leftarrow m_{J}\right) & \overset{C_{M}}{\to } & p(M|do(M_{J}\leftarrow m_{J})) \end{array}$$

*
where $\beta_{*}$ denotes the push-forward operation applied to the probability $p(H|do(H_{I}\leftarrow h_{I}))$, and $J=\Pi^{-1}\left(I\right)$ is the pre-image of I under Π.
*


In other words, aligned causal abstractions extend block alignment by enforcing a
*
symmetrical
*
consistency condition over interventions when both SCMs $C_{H}$ and $C_{M}$ are known: interventions on M have analogues on H and vice versa, i.e., Equation (Equation 18) holds and is
*
commutative*. This becomes relevant in situations where the user cannot parse the effect of an intervention on $H_{i}$ on the input X, i.e., they do not have access to $p(X|do\left(H_{i}\leftarrow h_{i}\right))$, and they are left to validate the effects of their actions through β. In this case, leveraging on the SCM $C_{M}$, the user can check how the mirrored intervention on $M_{j}$ spreads in the machine representations, and compare it with the corresponding representations given by β when the intervention is propagated on the user’s factors $H_{-i}$.

Therefore, while a map β being aligned is a necessary condition, it is not sufficient to guarantee a successful
*
name transfer
*
if $C_{M}$ is highly dissimilar from $C_{H}$. We show this situation explicitly where, despite having alignment between the user and the machine, the consistency condition in Equation (Equation 18) does not hold.

**Example 3.** *
We consider two SCMs, one over user variables $C_{H}$ and one over the machine ones $C_{M}$. As shown in Figure 7 (left), the two SCMs have a different structure and for ease of reference we refer to $H_{1}$ and $M_{1}$ as the* temperature *variable and to $H_{2}$ and $M_{2}$ as the *color* variable. Despite the different structure, we suppose $M_{1}$ and $M_{2}$ are aligned to $H_{1}$ and $H_{2}$, respectively, via an aligned map β. We indicate the overall causal graph as $C_{H\to M}$; see Figure 7 (right)*.*We can now check that $C_{M}$ is not an aligned abstraction of $C_{H}$ under β. In fact, intervening on $H_{1}$ leads to different results on $C_{M}$ and $C_{H\to M}$. For the former, changing the *temperature* amounts to modifying only the corresponding variable $M_{1}$ and does not affect $M_{2}$, as evident in Figure 7 (left). Conversely, a change in the* temperature*under alignment corresponds also to a change in* color*for the variable $M_{2}$, as depicted in Figure 7 (right). The two interventional effects, hence, do not coincide and $C_{M}$ is not an aligned causal abstraction of H.
*

## 5. Discussion and Limitations

Our work provides a crisp requirement that machine representations should satisfy to ensure interpretability, namely
*
alignment
*
with the human’s concept vocabulary. Next, we address important issues arising from this requirement.

### 5.1. Is Perfect Alignment Sufficient and Necessary?

It is natural to ask whether perfect alignment is a
*
sufficient
*
and
*
necessary
*
condition for interpretability of machine concepts. Recall that alignment is born out of two desiderata. The first one is that of
*
subjectivity*: a concept is understandable
*
to
*
a particular human observer, with different observers having different expertise and knowledge. This is captured by the human’s vocabulary H in our definition. The second one is that of guaranteeing that
*
machine and human concepts sharing the same name also share the same semantics*, translated into the desideratum that whenever a human concept changes the human can anticipate how this will change the machine representation. For instance, if the human and the machine see the same picture of a dog, the 
*human* can easily figure out what concept encodes the notion of “dog” and how it would change if they were to delete the dog from the picture. This last point takes into account, at least partially, the limited cognitive processing abilities of human agents.

*Is alignment sufficient?*
Simply ensuring that two agents share aligned representations does not automatically entail that symbolic communication will be successful. For instance, a human observer may misinterpret a machine explanation built out of aligned concepts simply due to inattention, confusion, or information overload. These are all important elements in the equation of interpretability, and we do not intend to dismiss them. The way in which information is *presented*
is about as important as the *contents* of the information being conveyed. The problem of designing interfaces that ensure the presentation is engaging and easy to understand is, however, beyond the scope of this paper. This does not impact our core message, that is, that *lack* of alignment can severely hamper communication and that therefore approaches for learning and evaluating conceptual representations should be designed with this requirement in mind.

*Is alignment necessary?* We also point out that perfect alignment is not strictly *necessary*, for two reasons. First, it is enough that alignment holds only *approximately*. Slight differences in semantics between machine and human concepts are unlikely to have major effects on communication. This is compatible with the empirical observation that people can often successfully communicate even without fully agreeing on the semantics of the words they exchange [63]. In practice, the degree of misalignment, and its impact of the communication, can be defined and measured, at which point the maximum allowed misalignment becomes an application-specific variable. Second, it may not be necessary that alignment holds *everywhere*. If two agents exchange only a subset of possible messages (e.g., explanations), concepts not appearing in those messages need not be aligned. For instance, ensuring a CBM classifying apples as ripe
or not to be interpretable only requires the concepts appearing in its explanations to be aligned, and possibly only those values that actually occur in the explanations (e.g., color=red but not color=blue). This can be understood as a more lax form of alignment applying only to a certain subset of (values of) the generative factors $g_{I}$, e.g., those related to apples. It is straightforward to relax Definition 6 in this sense by restricting it to a subset of the support of $p^{*}\left(G_{I}\right)$ from which the inputs X are generated, as these constrain the messages that the two agents can exchange.

### 5.2. Measuring Alignment

While there exist several metrics for measuring interpretability of concepts (discussed in Section 6.4), here we are concerned with techniques for assessing
*
alignment*.

Considering the relation between alignment and disentanglement (**D1**), one option is to leverage one of the many measures of disentanglement proposed in the literature [64]. The main issues is that most of them provide little information about how simple the map α (**D2**) is and, as such, they cannot be reused as-is. However, for the disentangled case (see Section 4.1), Marconato et al. [19] noted that one can measure alignment using the linear DCI [40]. Essentially, this metric checks whether there exists a
*
linear regressor
*
that, given $m_{J}$, can predict $g_{I}$ with high accuracy, such that each $M_{j}$ is predictive for at most one $G_{i}$. In practice, doing so involves collecting a set of annotated pairs $\left\{\left(m_{J},g_{I}\right)\right\}$, where the $m_{j}$’s and $g_{i}$’s are rescaled in [0,1], and fitting a linear regressor on top of them using $L_{1}$ regularization. DCI then considers the (absolute values of the) regressor coefficients $B\in R^{\left|J|\times |I\right|}$ and evaluates average dispersion of $B_{j:}$ for each machine representation $M_{j}$. In short, if each $M_{j}$ predicts only a single $G_{i}$, and with high accuracy, then linear DCI is maximal. The key insight is that the existence of such a linear map implies both disentanglement (**D1**)
*
and
*
monotonicity (**D2**), and therefore also alignment. The main downside is that the converse does not hold, that is, linear DCI cannot account for non-linear monotonic relationships.

The alternative we advocate is that of
*
decoupling
*
the measurement of **D1** and **D2**, and to leverage causal notions for the former. **D1** can, for instance, be measured using the
*
interventional robustness score
*
(IRS) [41], an empirical version of EMPIDA (Definition 2) that, essentially, measures the average effect of interventions on $G_{I}$ on the machine representation. Alternatives include, for instance, DCI-ES [65], which can better capture the degree by which factors are mixed and the mutual information gap (MIG) [66]. These metrics allow to establish an empirical map π between indices of the human and machine representations, using which it is possible to evaluate D2 separately. One option is that of evaluating Spearman’s rank correlation between the distances:
(19)$$|g_{i}-g_{i}^{\prime }{|}^{2}\;\mathrm{and}\;{∥E\left[M_{J}|do\left(G_{i}\leftarrow g_{i}\right)\right]-E\left[M_{J}|do\left(G_{i}\leftarrow g_{i}^{\prime }\right)\right]∥}_{2}^{2}$$
for interventions $g_{i}$ and $g_{i}^{\prime }$, leaving $G_{-i}$ fixed, for each i∈I and multiple traversals $\left(g_{i},g_{i}^{\prime }\right)$.

Unfortunately, none of the existing metrics are suited for non-disentangled generative factors $G_{I}$ or human representations H, which are central for alignment in the block-wise (Section 4.4) and general (Section 4.5) cases. Moreover, *alignment* and *block-alignment* share the computational cost and complexity of other disentanglement metrics [41,65,66], since both **D1** and **D2** can be adapted from them. The number of total concept combinations, even in the disentangled case (Definition 1), grows exponentially with the number of concepts *k*, which requires in practice bounds for the estimation. This is also a noteworthy problem in the disentanglement literature; see, e.g., [64]. We leave an in-depth study of more generally applicable metrics to future work.

### 5.3. Consequences for Concept-Based Explainers

Recall that CBEs explain the predictions of black-box models by extracting interpretable concepts $\hat{H}$ from the model’s internal representation M and then evaluating their contribution to the prediction (see Section 3.1). In this case, the requirement is that $\hat{H}$ is aligned to the human’s concept vocabulary H, irrespective of how the former is extracted. Notice that
*
alignment
*
is orthogonal to
*
faithfulness*, in the sense that an aligned representation can be unfaithful to the model, and a faithful representation misaligned with the human. In other words,
*
alignment is a property of the map from H to $\hat{H}$, while faithfulness is a property of the map between M and $\hat{H}$*. Evaluating faithfulness can be performed, e.g., via TCAV scores and assessing the degree of linear separation of concepts in the machine representation M. Another approach [36] measures the degree to which M can be reconducted to a causal decision graph on top of the concepts of interest. Other methods are discussed in Section 6.2.

If the mapping from M to $\hat{H}$ is
*
invertible*, then it is always possible map back and forth, in a lossless manner, from the machine representations M to the surrogate $\hat{H}$. This is a solid basis for faithfulness: whatever information is conveyed by an explanation built on $\hat{H}$ can always be cast in terms of the machine representation itself, and that whatever relation the latter has with the prediction can be mapped in terms of human concepts. The resulting explanation may no longer be simple or understandable, but it still contains all the information of the original message.

In the general case, however, it is non-trivial to find a suitable invertible function. Suppose the user provides the machine with annotated examples $\left(x_{i},h_{i}\right)$ and that these are used (as is common with supervised CBEs; see Section 6.2) to learn the mapping from M to $\hat{H}$. Ensuring that this is invertible requires potentially an enormous amount of examples. To see this, consider a simple case in which the human concepts H are binary and disentangled and that M and H are related by a (possibly complex) invertible mapping that is not an alignment. Even in this ideal case, it might take up to $2^{\ell }$ examples, where *ℓ* is the dimension of H, to align the two representations, as this involves finding the correct permutation from M to H. Alignment can help in this regard. In fact, if M
*
is
*
aligned to H, the number of required examples scales as O(ℓ), because a single intervention to each user concept $H_{i}$ is sufficient to find the corresponding aligned element $M_{j}$.

In summary, not only do unaligned (black-box) models imply CBEs require more supervision on the user concepts to acquire a invertible transformation ensuring faithfulness, but it is also likely that the representation M mixes together the interpretable factors $G_{I}$ with the non-interpretable ones $G_{-I}$, making it more difficult to extract a concepts $\hat{H}$ aligned to H.

### 5.4. Consequences for Concept-Based Models

As discussed in Section 6.1, most CBMs acquire concepts using a variety of heuristics that do not guarantee alignment. To the best of our knowledge, GlanceNets [19] are the only CBM that
*
explicitly
*
optimizes for alignment, and as such avoids concept leakage. They do so by combining a variational auto-encoder mapping from the input X to a machine representation $M=(M_{J},M_{-J})$ where only the first partition is used for prediction. These are computed using a simple linear layer, as is customary. The variational-auto encoder is trained with some amount of concept-level annotations. This encourages both disentanglement [67]
*
and
*
monotonicity, and hence alignment, for
*
in-distribution
*
data. In turn, this also prevents concept leakage. In order to avoid leakage for
*
out-of-distribution
*
data, GlanceNets also implement an
*
open-set recognition
*
step [57]. This is responsible for detecting inputs encoding concepts that have never been observed during training. Whenever these are detected, GlanceNets refuse to output a prediction for them, thus avoiding leakage altogether.

From our perspective, GlanceNets have two major downsides. First, they are designed to seek alignment with respect to the generative factors underlying the observations. As we argued, however, interpretability requires alignment with respect to the human’s concept vocabulary. Second, GlanceNets require a moderate but non-trivial number of annotations. How to acquire them from the human observer remains an open problem, as discussed in Section 5.5.

Summarizing, GlanceNets could be repurposed for solving alignment in the disentangled case discussed in Section 4.1 by combining them with a suitable annotation elicitation procedure. They are, however, insufficient to solve disentanglement when the ground-truth concepts are not disentangled, and new solutions will be necessary to tackle these more complex and realistic settings.

### 5.5. Collecting Human Annotations

Both metrics and learning strategies for alignment require some amount of annotations for the human factors H. This is a core requirement related to the subjective nature of interpretability. One option is that of distributing the annotation effort among crowd-workers which, however, is impractical for prediction tasks that require specific types of expertise, like medical diagnosis. An alternative is that of gathering together annotations from different online resources of large language models [68]. Doing so, however, can lead to a lack of completeness (a necessary concept might be missing) and ambiguity (concepts annotations might mix together different views or meanings). This kind of supervision cannot guarantee alignment to a specific human observer.

Reducing the annotation effort for personalized supervision is challenging. Under the assumption that of leveraging generic concept annotations obtained using the above methods to pre-train the concept extractor, and then fine-tune the resulting model using a small amount of personalized annotations. This strategy can save annotation effort as long as the generic annotations contain most of the information necessary to retrieve the observer’s concepts. An alternative is to leverage concept-level interactive learning [69,70], to request annotations only for those concepts that are less
*
aligned*. This is of particular interest for interventions at the concept in CBMs [71,72]. It was also shown that interactive strategies can increase the amount of *disentanglement* during the learning phase [73]. However, how to collect interventional concepts remains an open challenge. Naturally, one might also consider combining these two strategies, that is, interleaving fine-tuning with interactive learning, for additional gains. How to estimate alignment (or some lower bound thereof) in the absence of full concept annotations is, however, an open research question and left to future work.

## 6. Related Work

While concepts lie at the heart of AI [74], the problem of acquiring
*
interpretabile
*
concepts has historically been neglected in representation learning [32]. Recently, concepts have regained popularity in many areas of research, including explainable AI [1], neuro-symbolic AI [75,76], and causality [31,38], yet most concept acquisition strategies developed in these areas are only concerned with task accuracy, rather than interpretability. The presence of users in the framework establishes a connection between our work and cognitive sciences [77,78]. For our purposes, we model the user concepts with variables H belonging to their internal structure, whereas other issues arising from the nature of explanations and the subjectivity content are not taken into consideration. This does not exclude that further studies in cognitive science will strengthen the notion of alignment pertaining to individuals’ limitations (see, e.g., [79]). Our work builds on causal representation learning [31] that offers a solid basis for capturing some aspects of the mutual understanding between the human and machine, rooted in the definition of alignment.

Next, we briefly overview strategies for acquiring interpretable representations and highlight their shortcomings for properly solving human-interpretable representation learning.

### 6.1. Unsupervised Approaches

A first group of strategies learn concepts directly from unlabeled data. Well-known theoretical results in deep latent variable models cast doubts on the possibility of acquiring representations satisfying
*
any
*
property of interest, including
*
disentanglement
*
and
*
interpretability*, in a fully unsupervised manner in absence of a strong architectural bias [67,80]. This stems from the fact that, as long as the concept extraction layers are “flexible enough” (i.e., have no strong architectural bias), predictors relying interpretable and uninterpretable concepts can achieve the very same accuracy (or likelihood) on both the training and test sets. As a consequence,
*
unsupervised strategies that only maximize for accuracy cannot guarantee interpretability unless they are guided by an appropriate bias*. The main challenge is determining what this bias should be.

Several, mutually incompatible alternatives have been proposed. Unsupervised CBEs discover concepts in the space of neuron activations of a target model. One common bias is that concepts can be retrieved by performing a
*
linear decomposition
*
of the machine’s representation [48]. Specific techniques include k-means [13], principal component analysis [81], and non-negative matrix factorization [14,15]. Concept responsibility is then established via feature attribution methods.

Two common biases used in CBMs are
*
sparsity
*
and
*
orthonormality*. Self-Explainable Neural Networks [16] encourage the former by pairing an autoencoder architecture for extracting concepts from the input together with a (simulatable [21]) task-specific prediction head, and then combining a cross-entropy loss with a penalty term encouraging concepts to have sparse activation patterns. Concept Whitening [27] implements a special bottleneck layer that ensures learned concepts are
*
orthogonal*, so as to minimize mutual information between them and facilitate acquiring concepts with disjoint semantics, as well as
*
normalized
*
within comparable activation ranges. The relationship between sparsity, orthonormality, and interpretability is, however, unclear.

Based on the observation that humans tend to reason in terms of concrete past cases [4], other CBMs constrain concepts to capture salient training examples or parts thereof, i.e.,
*
prototypes*. Methods in this group include Prototype Classification Networks [82], Part-Prototype Networks [17], and many others [83,84,85,86]. At a high level, they all memorize one or more prototypes (i.e., points in latent space) that match training examples of their associated class only. Predictions are based on the presence or absence of a match with the learned prototypes. The interpretability of this setup has however been called into question [25,26]. The key issue is the matching step, which is carried out in latent space. The latter is generally underconstrained, meaning that prototypes can end up matching parts of training examples that carry no useful semantics (e.g., arbitrary combinations of foreground and background), as long as doing so yields high training accuracy.

None of these approaches takes the human’s own concept vocabulary H into account.

### 6.2. Supervised Strategies

A second family of approaches leverages concept
*
annotations
*
(or some form of weak supervision). Among supervised CBEs, Net2vec [12] defines linear combinations of convolutional filters, and fits a linear model to decide whether their denoised saliency maps encode a given concept or not, yielding a binary segmentation mask. TCAV [11] defines concepts as directions, or concept-activation vectors (CAVs), in latent space. These are obtained by adapting the parameters of per-concept linear classifiers trained on a separate densely annotated data set to the machine’s embedding space. Concept attributions are proportional to the degree by which changing their activations affects the prediction. Zhou et al. [87] also relies on CAVs, but computes explanation by solving an optimization problem. A second group of supervised CBEs makes use of non-linear maps instead [88,89,90]. For instance, CME [88] uses all activations of the model to learn categorical concepts via semi-supervised multi-task learning, while INN [90] fits a normalizing flow from the machine representation to the concepts so as to guarantee their relationship is bijective. In CBEs, it is also important to estimate the faithfulness of the concepts. TCAV measures the degree of linear separation among concepts [11], Yeh et al. [91] introduced the *completeness score* to evaluate the amount of information the concepts contain about the label, and Fel et al. [48] considered also the CBE *stability*, *fidelity*, and *out-of-distribution discrepancy*.

Supervised CBMs like Concept-Bottleneck Models [18], Concept Whitening [27], and GlanceNets [19], among others [47,92,93] define a loss training penalty, for instance a cross-entropy loss, encouraging the extracted concepts to predict the annotations. Recently, these methods have been extended also to graph neural networks [94,95]. This solution seems straightforward: there is no more direct way than concept supervision to guide the model toward acquiring representations with the intended semantics. It also circumvents the negative theoretical results outlined in Section 6.1.

However, models that accurately match the supervision do not necessarily satisfy content-style separation or allow to have disentangled representations, which, as discussed in Section 4.3, would lead to a non-negligible amount of
*
concept leakage
*
[28,29]. In contrast, alignment explicitly takes both properties into account. Another major issue is the supervision itself, which is frequently obtained from general sources rather than from the human observer themselves, meaning the learned concepts may not be aligned to the concept vocabulary of the latter. Two notable exceptions are the interactive concept learning approaches of Lage and Doshi-Velez [69] and of Erculiani et al. [96], which are, however, unconcerned with concept leakage.

To the best of our knowledge, GlanceNets [19] are the only CBM that explicitly optimizes for alignment, and as such, avoid leakage, yet they do so with respect to generative factors rather than human concepts. As discussed in Section 5.4, however, GlanceNets can in principle be adapted to solve human-interpretable representation learning by combining them with a suitable annotation acquisition strategy. We plan to pursue this possibility in future work.

### 6.3. Disentanglement

Another relevant area of research is that on learning disentangled representations. Here, the goal is to uncover “meaningful”, independent factors of variation underlying the data [33,67,97], with the hope that these are also interpretable [32]. Most current learning strategies rely on extensions of variational auto-encoders (VAEs) [66,97,98,99,100,101]. As anticipated in Section 6.1, unless suitable architectural bias is provided, unsupervised learning cannot guarantee the learned representations are disentangled. Motivated by this, follow-up works seek disentanglement via either concept supervision [102], weak supervision [103,104], and other techniques [73,105,106]. Disentanglement, however, is unconcerned with the human’s concept vocabulary, and furthermore it is weaker than alignment, in that is does not readily support name transfer.

Independent component analysis (ICA) also seeks to acquire independent factors of variation [107,108,109]. These assume the generative factors are independent from each other and determine an observation via an injective or invertible map. The objective of ICA is to recover the generative factors from the observations. While the linear case is well understood [107], the non-linear case is arguably more difficult. It was shown that
*
identifying
*
the ground truth factor is impossible in the unsupervised setting [110]. This is analogous to the results mentioned in Section 6.1 and, in fact, a formal link between deep latent variable models and identifiability has recently been established [80]. On the positive side, it is possible to show that providing auxiliary supervision on the factors guarantees identification up to permutation and negation, a property known as
*
strong identifiability*. *Weak identifiability* [111] relaxes it, whereby the generative factors are recovered up to a transformation of the form Ag+b, where rank(A)≥min(dimG,dimM) and M is the machine representation and b is an offset. Hyvarinen and Morioka [58] also contemplate
*
identifiability up to element-wise non-linearities*, that is, given by the class of transformations Aσ[g]+b, where σ can be a non-linear. If σ is restricted to be monotonic and *A* is an element-wise transformation, according to condition D1 in Definition 6, then this form of identifiability matches that of alignment in the disentangled case. However, this formulation refers to identification of the generative factors, while alignment is defined specifically in terms of human concepts. Moreover, we do not assume to the map from human to machine concepts to be injective, nor to be exact.

### 6.4. Metrics of Concept Quality

Several metrics have been proposed for assessing the quality of extracted concepts and of explanations built on them. Standard measures include accuracy and surrogates thereof [11]; Jaccard similarity [12]; sparsity, stability and ability to reconstruct the model’s internal representation [48]; and the degree by which concepts constitute a sufficient statistics for the prediction [91]. We refer to [22] for an overview. These metrics, however, either entirely neglect the role of the human observer, in that concept annotations are either not used or not obtained from the observer themselves, or fail to account for disentanglement and concept leakage. Alignment fills these gaps. Recently, two new metrics have been proposed to measure the concept impurity across individual learned concepts and among sets of representations [112], but the relation with alignment has not been uncovered yet.

There also exist a number of metrics for measuring disentanglement, such as β-VAE score [97], Factor-VAE score [99], mutual information gap [66], DCI [40], and IRS [41]. DCI provides also information about the informativeness of its estimate, and, following [19], it can be repurposed to measure a form of alignment where the μ transformations are linear Definition 6. Suter et al. [41] propose EMPIDA to analyze disentanglement from a causal perspective, upon which we base the construction of alignment. As mentioned in Section 5.2, these metrics can be used to evaluate **D1** in the definition of alignment, and therefore alignment itself when paired with a metric for measuring the complexity of α (**D2**). Their properties are extensively discussed in [64].

### 6.5. Neuro-Symbolic Architectures

The decomposition between low-level perception—that is, mapping inputs to concepts, also known as
*neural predicates
* in this setting—and high-level inference outlined in Section 3.1 applies also to many neuro-symbolic (NeSy) models. Examples include DeepProblog [113], Logic Tensor Networks [114], and related architectures [115,116,117,118,119,120,121,122,123,124,125]. The biggest differences between CBMs and NeSy architectures is how they implement the top layer: the former rely on simulatable layers, while the latter on reasoning layers that take prior symbolic knowledge into account and are not necessarily simulatable.

Recent works [126,127] showed that learning a NeSy model consistent with prior knowledge using only label supervision is insufficient to guarantee the neural predicates capture the intended semantics. For instance, it is not uncommon that NeSy architectures attain high prediction accuracy by acquiring neural predicates that encode information about distinct and unrelated concepts. Interpretability of the
*neural predicates*, however, also requires alignment, meaning that our results apply to these NeSy architectures as well.

## 7. Conclusions

Motivated by the growing importance of interpretable representations for both post hoc and ante-hoc explainability, we have introduced and studied the problem of *human-interpretable representation learning*. Our key intuition is that concepts are interpretable only as long as they support symbolic communication with an interested human observer. Based on this, we developed a formal notion of alignment between distributions, rooted in causality, that ensures concepts can support symbolic communication and that applies to both post hoc concept-based explainers and concept-based models. In addition, we clarified the relationship between alignment and the well-known notions of disentanglement, illustrating why the latter is not enough for interpretability, and uncovered a previously unknown link between alignment and concept leakage. Finally, looking at alignment in the most general case, we also unearthed its link to causal abstractions, which further cements the link between interpretability and causality and that we plan to expand on in future work. With this paper, our aim is that of bridging the gap between the human and the algorithmic sides of interpretability, with the hope of providing a solid, mathematical ground on which new research on human-interpretable representation learning can build.

## Figures and Tables

**Figure 1 entropy-25-01574-f001:**
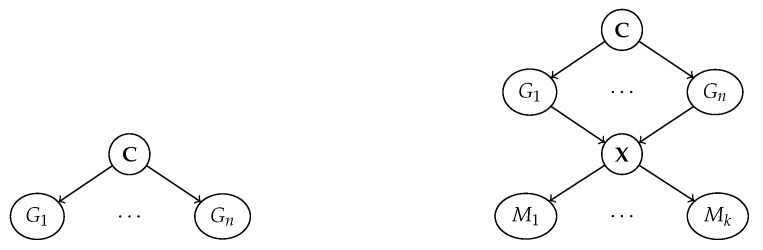
SCMs illustrating two different notions of disentanglement. *Left*: the variables G =$\left\{G_{1},\ldots ,G_{n}\right\}$ are disentangled. *Right*: Typical data generation and encoding process used in deep latent variable models. The machine representation M = $\left\{M_{1},\ldots ,M_{k}\right\}$ is *disentangled with respect to* the generative factors G if and only if each $M_{j}$ encodes information about one $G_{i}$ at most.

**Figure 2 entropy-25-01574-f002:**
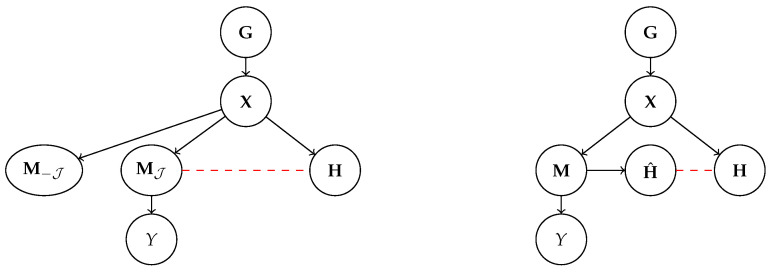
**Left**: following the generative process p(X∣G), *concept-based models* (CBMs) extract a machine representation $M=(M_{J},M_{-J})$ via $p_{\theta }\left(M|X\right)$, of which only $M_{J}$ is used to predict the label *Y*. $M_{J}$ contains all interpretable concepts, and as such it has to be *aligned* to their user concepts H (Section 4): the corresponding map is reported in red. **Right**: generative process followed by *concept-based explainers* (CBEs). Here, the machine representation M is *not* required to be interpretable. Rather, the concept-based explainer maps it to extracted concepts $\hat{H}$ and then infers how these contribute to the prediction *Y*. Here, alignment should hold between $\hat{H}$ and H.

**Figure 3 entropy-25-01574-f003:**
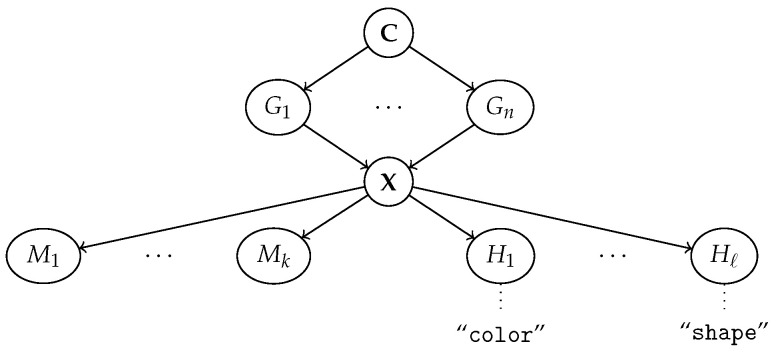
Graphical model of our data generation process. In words, *n* (correlated) generative factors exist in the world G = $\left(G_{1},\ldots ,G_{n}\right)$ that *cause* an observed input X. The machine maps these to an internal representation M = $\left(M_{1},\ldots ,M_{k}\right)$, while the human observer maps them to its own internal concept vocabulary H = $4(H_{1},\ldots ,H_{\ell })$. Notice that the observer’s concepts H may, and often do, differ from the ground-truth factors G. The concepts H are what the human can understand and attach names to, e.g., the “color” and “shape” of an object appearing in X. The association between names and human concepts is denoted by dotted lines. We postulate that communication is possible if the machine and the human representations are *aligned* according to Definition 6.

**Figure 4 entropy-25-01574-f004:**
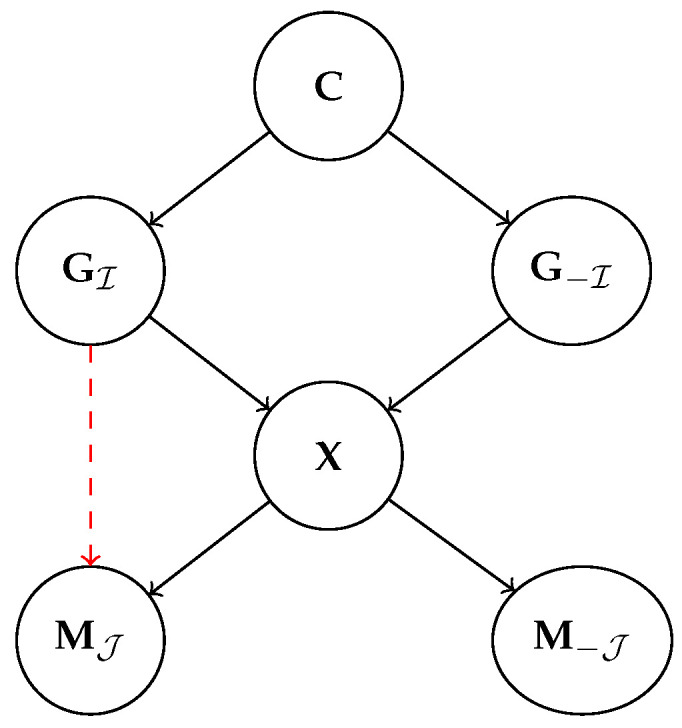
Simplified generative process with a single observer, adapted from [19]. Here, C is unobserved confounding variables influencing the generative factors G, and M is the latent representation learned by the machine. The red arrow represents the map α.

**Figure 5 entropy-25-01574-f005:**
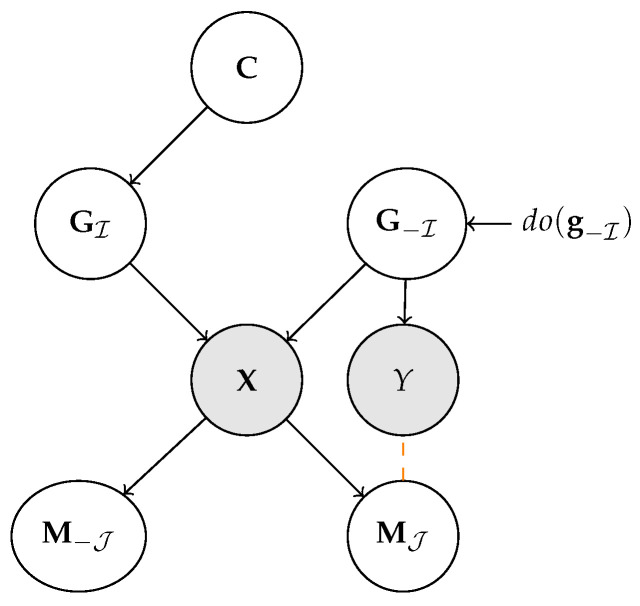
Generative process for Concept Leakage. A predictor observes examples (X,Y) and infers *Y* from its interpretable representation $M_{J}$ using a learnable conditional distribution $q_{\lambda }\left(Y|m_{J}\right)$, indicated in orange. Since the label *Y* depends solely on $G_{-I}$, we would expect that it
*cannot* be predicted better than at random: intuitively, if this occurs it means that information from $G_{-I}$ has leaked into the interpretable concepts $M_{J}$. Any intervention $do(G_{-I}\leftarrow g_{-I})$ on the uninterpretable/unobserved concepts detaches these from C, meaning that the label truly only depends on $G_{-I}$.

**Figure 6 entropy-25-01574-f006:**
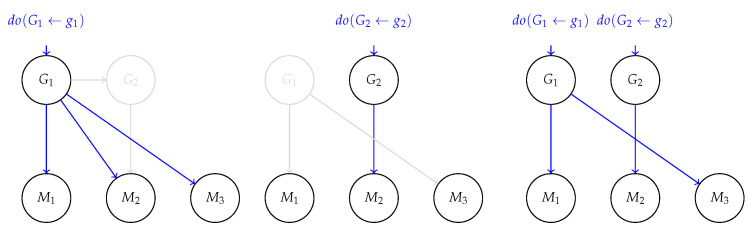
Block-aligned representation when $C_{G}$ has causal connections. (**left**) An intervention on $G_{1}$ affects all representation (displayed in blue), since $\left(M_{1},M_{3}\right)$ are block-aligned to $G_{1}$ and $M_{2}$ is aligned to $G_{2}$. (**center**) Conversely, an intervention on $G_{2}$ only affects $M_{2}$, leaving the remaining representations untouched. (**right**) Intervening on all G has the effect of isolating the corresponding aligned representations from other interventions. In this case, intervening on $G_{2}$ removes the causal connection with $G_{1}$, so that $M_{2}$ does not depend on the intervention of $G_{1}$. Refer to Example 2 for further details.

**Figure 7 entropy-25-01574-f007:**
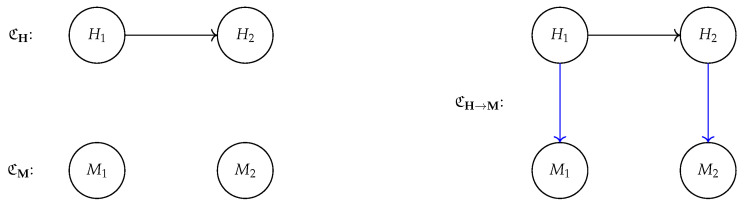
Absence of aligned causal abstraction. (**left**) The user’s $C_{H}$ incorporates a causal connection between $H_{1}$ to $H_{2}$, while the machine one $C_{M}$ presents no causal connections. (**right**) The total SCM $C_{H\to M}$ of user’s and machine’s concepts resulting from an **aligned** map β:H→M (in blue). Refer to Example 3 for further discussion.

## Data Availability

No new data were created or analyzed in this study. Data sharing is not applicable to this article.

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
