# Peer review of "Interpretability Is in the Mind of the Beholder: A Causal Framework for Human-Interpretable Representation Learning"

_entropy, 2023, doi:10.3390/e25121574_

Round 1
Reviewer 1 Report
Comments and Suggestions for Authors
The authors propose a mathematical framework to study the problem of human-interpretable representation learning for both post-hoc explainers and concept-based neural networks. I'm not sure I have well understood the paper's idea and method, but I feel it's an interesting topic. I think the manuscript should be accepted after some revisions.
1. When the abbreviation first appears, the full name should be given, such as AI in Line 24.
2. The structure of the paper is not very reasonable and clear. In general, it is divided into Introduction, Method, Results and discussion, Conclusion, in which the Related work follows the Introduction.
Comments on the Quality of English LanguageNo comments.
Author Response
We thank the reviewer for their honest feedback. In summary, our paper provides a mathematical definition of interpretability applicable to concepts learned by a machine from data. Our definition is grounded on the intuition that humans can understand these concepts if and only if they can associate the right name to them (by playing a “name assignment game”, which we also define). Despite how intuitive this idea is, it is also very novel and it leads to rather profound consequences about how concepts should be learned from data and about how their interpretability should be measured. Our insights also cast doubts on existing techniques and measures for concept acquisition.
Concerning your feedback, we made sure all acronyms were properly defined before their first usage. We have chosen to leave the related work in Section 6, partly because all necessary concepts are introduced in Section 2, and partly to keep the methodological discussion close to the motivation (found in the Introduction).
Reviewer 2 Report
Comments and Suggestions for Authors
This is a research work about the invention of a framework that captures causal “language”, meaning through causal models some basic characteristics of concept and representation learning. The researchers start with the creation of causal models expressing both human and machine internal representations of concepts and then elaborate on their alignment and misalignment as well as the sufficient conditions thereof. Overall in the paper, several small examples describe how this causal framework can be used, and some preliminary ideas about how interpretability is achieved are also there. Concept-learning methods are also analyzed from the lens of this framework which provides a general guideline for concept-based learning models.
The paper does contain significant content to justify a publication. The whole framework is novel and the paper goes extensively through the theoretical (mathematical) basis. Some more elaborate examples are necessary though.
Missing background and related work contains two important publications: on the one hand a concept explainer of the state-of-the-art Graph Neural Networks (GNN) architecture is missing:
- Magister, L. C., Kazhdan, D., Singh, V., & Liò, P. (2021). Gcexplainer: Human-in-the-loop concept-based explanations for graph neural networks. arXiv preprint arXiv:2107.11889.
https://doi.org/10.48550/arXiv.2107.11889
Another important work for concept learning characteristics and metrics is the following:
- Holzinger, A., Saranti, A., Angerschmid, A., Finzel, B., Schmid, U., & Mueller, H. (2023). Toward human-level concept learning: Pattern benchmarking for AI algorithms. Patterns.
https://doi.org/10.1016/j.patter.2023.100788
The concept of alignment and human expectations is also tackled in the following publication (not a must though):
- Finzel, B., Saranti, A., Angerschmid, A., Tafler, D., Pfeifer, B., \& Holzinger, A. (2022). Generating explanations for conceptual validation of graph neural networks. KI-Künstliche Intelligenz, 1-15.
doi: 10.1007/s13218-022-00781-7
The claim in lines 524-525 that semantic compatibility is a psychological issue needs a reference.
The methods are clearly described overall, but there are still some open issues. In line 98 – what is sufficient then? On page 6 the concrete relationship between M_G and M_I needs to be more concrete. Figure 2 and its description is difficult to follow. How is the distribution in line 222 implemented by a Neural Network concretely? Why is the MAP inference chosen on page 6, there are other types as well. What is the difference between relevance (as mentioned by most xAI methods) and responsibility on page 7? In line 238 it is not required to be interpretable but can it also be interpretable? In lines 250-251: when does it happen, and how can it be detected? In line 289 yes, the concepts H are understandable by definition but differently for each human. How can one make sure the alignment in lines 294-295? In the caption of Figure 3, is communication impossible if not aligned? How are the authors on page 9 sure that they will associate (line 319)? In Definition 6 it is not explained here how the noise variable will play a role. In lines 346 – 356, it would be preferable to have a concrete example to show “multiple” interpretability. Lines 389 – 395 need more elaboration to be understandable. Why aligned representations cannot suffer from concept leakage is not clear. Lines 478-490 need more concrete examples. How realistic is it that only a single Mj’ is aligned to G_k? Lines 517-536 are also not understandable.
Lines 588-589 need an example. The orthogonality of alignment and faithfulness can exist but not necessarily. How is the mapping going to be created in line 746 is the question.
The paper is very well-written and has a clear structure. There are no typos found apart from „simulatable“ in line 521.
Author Response
We thank the reviewer for their suggestions and for finding the paper clear and interesting. Let us now address their feedback in detail:
- Missing related works: Thank you for pointing out these works, they are definitely relevant and we have included them in the related work section (section 6.2). We would like to point out that our analysis is model agnostic, and as such it does apply to relational data (e.g., graphs) as well.
- References about the psychological nature of semantic compatibility: Thank you for pointing this out. We have reworded the text to avoid this claim entirely, as it is not central to our message.
- Line 98: We simply wanted to remark that alignment combines two properties: one is disentanglement and the other is monotonicity (invertibility). (In the case of block-alignment, the latter translates to invertibility and simulatability). We did rephrase the text to make this more clear, see section 1.2.
- Exemplify the relationship between M_G and M_I has to be clearer. The vector G contains the ground-truth factors, that is, the true color and shape of a 2D sprite provided as input to the model. The vector M is the representation that the model has learned for the same 2D sprite.
The machine representation M can be split in two parts: M_J, which contains the concepts proper, that is, representations that ought to be interpretable, and M_{\not J}, which contains representations that are not constrained to be interpretable. This split ensures our conceptualization can capture models (like concept-bottleneck models) in which all representations are interpretable (that is, M_{\not J} is empty), as well as GlanceNets and other models, which do make use of additional uninterpretable representations (M_{\not J} is not empty). The key point of Figure 2 is that only M_J has to be aligned to the user’s concept vocabulary H. The only relationship between M_J and M_{\not J} is that they are conditionally independent given the input X, which is the case when they are computed by any feed-forward neural network (as is the case for all models we consider).
- The caption of Figure 2 is not clear enough: Thank you. We clarified the caption.
- Implementation of p(M | X): the concept extractor is often a feed-forward neural network (e.g., a multi-label CNN when dealing with binary concepts, like objects or parts thereof, or as a variational auto-encoder when dealing with continuous concepts, like colors and angles). Recently, it has also been implemented using large language models. We added a footnote to clarify this, see Section 3. Notice that our results hold regardless.
- Why choose MAP: We acknowledge that also other inference methods can be applied, although it is common practice to make use of MAP for inference and prediction in concept-based models. We have clarified this point. Again, our results do not hinge on the specific implementation.
- Responsibility versus relevance: We used the two terms interchangeably. We clarified this in the introduction.
- Interpretabilty of M in CBEs: We interpret this question as follows: can the machine representation of a black box model be interpretable? The answer is that, in certain situations, this may be the case. The model may - out of luck or architectural bias - have learned to represent interpretable concepts. However, this is most often not the case, both because generally neural networks are trained for performance without any additional bias encouraging interpretability, and because “neurons have no name”: if a user doesn’t know that neuron 56 represents “redness”, then they cannot trivially interpret what the network thinks without first implementing a procedure analogous to our name transfer process or a post-hoc XAI step.
- The lack of faithfulness between M and H^: Consider TCAV. At a high level, this approach identifies a vector in embedding space for each concept in a vocabulary. The projection of an input point onto this vector is used to determine the degree to which this concept appears in that input. However, these vectors are learned from data, which means that they are (rough) approximations (for instance, if the data is noisy or scarce). Furthermore, this procedure assumes the same concept vector is constant across space, and therefore implicitly that the embedding space is linear. This is often not the case for deep neural networks. In both situations, the concept activations extracted by TCAV have no guarantee of capturing exactly all the information the model is using for its predictions. It is not clear to what extent this problem can be solved. This is precisely one of the main motivations for developing concept-based models, where this post-hoc processing is not needed. We added further comments on how to evaluate faithfulness in Section 6.2.
- H is understandable by definition only to the human to whom they belong: True, we clarified our claim, see section 3.3.
- How to check for alignment: Alignment can be evaluated based on the definition (6) and in section (5.2) we report practical ways to measure it.
- Is communication possible even without alignment? Alignment ensures all possible messages exchanged between the machine and the human can be understood by both parties, specifically because the two agents associate the same semantics to all the concepts used in those messages. It is true that if the set of messages being exchanged is limited, alignment needs only to hold for those concepts that appear in the messages. We added a forward pointer to Section 5, where we clarified this aspect. Lack of alignment means the semantics that the machine attaches to a concept (say, the concept of ``red’’) differs substantially from that attached by the human to the same concept. For instance, the machine says ``red’’, but they really mean ``blue’’ or ``for some inputs red, for other inputs purple’’. Hence, messages containing that concept will be difficult to interpret correctly by the receiving end.
- How does alignment ensure semantics transfer?: Alignment is specifically designed so that, when it holds, the ``name transfer’’ procedure described in Section 4 succeeds. In short, every time the human changes the input according to their own concepts H (e.g., they change the hue of the sprite in the input image), the machine representation - if aligned - changes in a way that is 1) compatible with the user’s change (the machine representation of hue, and only that, changes), and 2) simulatable (increasing human hue increases machine hue and vice versa: it is a simple, non-arbitrary transformation whose effects are easy for people to anticipate). Hence, by playing this game, a sufficiently motivated human can transfer names from each H_i to the corresponding M_j (or H^_j). This is precisely why alignment is defined as a combination of disentanglement (in support of 1) and monotonicity (in support of 2). Other factors contributing to the success of name transfer are outlined in Section 5. Notice that our name transfer procedure is purely hypothetical: we do not expect people to use it in full. However, unless we can guarantee that it succeeds, we cannot guarantee that the human and the machine agree on the semantics of all concepts. Alignment is a mathematical property that can be measured experimentally, and acts as a surrogate for the name transfer procedure.
- The role of noise in Def. 6: The original definition of alignment given by Marconato et al. in 2022 was deterministic. The issue with this is that it does not account for stochasticity in the learning process: in practice, the learned mapping from inputs to concepts is not exact. The additional noise terms account for this. Notice that, however, our definition requires that the distribution’s mean is still ``well behaved’’, that is, that it is a monotonic transformation. We have clarified this point in the text after definition 6.
- “multiple interpretations” for the remarks following definition 6: Thank you, we agree this paragraph was not clear. We have rewritten it. We merely wanted to state that the definition as it is applies to both CBMs and CBEs by replacing the variables that alignment applies to to those appearing in those two cases.
- The explanation of concept leakage is not clear: We amended this by discussing more in depth what is the impact of leakage after Example 1.
- Aligned representations do not suffer leakage: Definition 7 shows that concept leakage can be viewed as the degree by which the model’s concepts M_J depend on unwanted (or “stylistic”) factors G_{-I}. Proposition 2 provides upper and lower bounds for the mutual information between these two terms. If the mutual information is zero, the M_J and G_{-I} are independent, and thus no leakage is possible. Finally, Proposition 3 remarks that if M_J is aligned to G_I (the wanted, or “content” factors), since alignment entails content-style separation, it must be the case that M_J is independent of G_{-I}. Hence, their mutual information is zero, and concept leakage must be zero. We adapted this overview explanation in the text, see section 4.3.
- Concrete examples for leakage in-distribution: We have clarified the text right before Corollary 1.
- Multiple M_j’s can be aligned to a G_k: Our definitions of alignment and block-alignment allow for the same G_k to be encoded by multiple M_j’s. Doing so does not violate alignment.
- Semantic compatibility in not clear: Thank you for pointing this out, we have clarified the text, see section 4.4 after definition 8.
- Example on the interventional independence: We included a reference to the example in the main text to make the consequences of proposition 4 more intuitive.
- Orthogonality between faithfulness and alignment: Faithfulness holds for explanations that manage to capture the model’s reasoning, alignment holds for explanations whose concepts match the human’s vocabulary. Notice how faithfulness is about the machine, and alignment about the human observer. Hence, they are orthogonal. One has alignment without faithfulness if the explanations can be understood by the human but they lie about the machine’s reasoning. We have faithfulness without alignment when the explanations fit the machine’s reasoning perfectly but are completely uninterpretable. For CBEs, faithfulness of explanations is typically assessed by the degree to which the concepts of the vocabulary are encoded in the representation of the model. Having a faithful explanation, however, is insufficient since the concepts may not be aligned. One can then check the relation between H^ and H, and check for (block-) alignment of H^. Ultimately, both faithfulness and alignment are necessary to evaluate the interepretability of the model.
Reviewer 3 Report
Comments and Suggestions for Authors
The authors propose a mathematical framework for acquiring interpretable representations, which is based on the notion of explicitly modeling human stakeholders as external observers. I believe the article is interesting and mathematically sound. The article proposed methodology is very interesting from the theoretical point of view, and the authors establish some non-trivial results on this formalization. However, I believe there are some minor and major issues that the authors would need to address:
- The authors claim that "focus in XAI has recently shifted toward explanations expressed in terms of higher-level symbolic representations [...]". I do not believe that this statement is entirely true, however, as most of the research in XAI is still mostly devoted to low-level "explanation" techniques, like feature (or pixel) attribution, or approaches that, while they can be understood as high-level (e.g., natural language explanations), cannot be easily mapped to what the authors called concepts.
- The authors' proposal to "view interpretability as the machine's ability to communicate with a specific human-in-the-loop" seems hardly novel, indeed it has been a major topic in the human-in-the-loop and XAI literature that ultimately could be traced back to the psychological (or cognitive) investigation of explanations, see e.g.
Miller, T. (2019). Explanation in artificial intelligence: Insights from the social sciences. Artificial intelligence, 267, 1-38.
Cabitza, F., Campagner, A., Malgieri, G., Natali, C., Schneeberger, D., Stoeger, K., & Holzinger, A. (2023). Quod erat demonstrandum?-Towards a typology of the concept of explanation for the design of explainable AI. Expert Systems with Applications, 213, 118888.
More generally, I believe that since the authors emphasize the human observer's role, it would be interesting to better connect their approach with the extensive literature on explanations arising from psychology, which explicitly studied the human aspect of this notion.
- Definition 5 does not seem to be a formalization of the notion of "Human-interpretable representation learning", as the notion "enables a machine and a specific human stakeholder to communicate using those representations" does not seem to have been formalized by the authors: what do you mean with "to communicate using those representations"?
- More in general, I am not fully convinced that the notions proposed by the authors can be really understood as an "operational" definition of interpretability. From the conceptual point of view, the notion of "enabling to communicate using those representations" seems hardly operationalizable within the framework proposed by the authors, simply because we are not usually able to access the human concepts: this implies that we would not generally be able to verify alignment. In Section 5.5 the authors propose to collect human annotations to address this problem, but I believe they should expand more on this aspect: indeed, while the authors claim that they want to put into the focus the "human-interpretable" notion, the human aspect seems to be relatively sidelined within the article.
- Also from the computational point of view, the authors do not provide indications about the complexity and scalability of the proposed approach. For example, in Section 5.2 the authors propose to measure D2 using Eq. 19: what is the complexity of evaluating 19? Also, since one normally would only consider a probabilistic approximation of this computation (as, in general, it may be impossible to evaluate the expected effect on M_I of performing the intervention G_i <- g_i), it would be useful if the authors provided some (probabilistic) guarantee.
In conclusion, I believe that this article has merit and should ultimately be published in this journal: I hope that the above suggestions could be useful to the authors for improving their article.
Author Response
We thank the reviewer for their critical feedback for appreciating the novelty and for their interest in our work. Below we answer the minor and major issues raised in the revision:
- Low-level explanations are still being studied in XAI. Good point, we were a bit too eager. We amended the abstract.
- The connection with cognitive science research. We agree that the subjective nature of explanations has been explored in the XAI and psychology literature. This is not what is novel about our proposal. What is novel is that we explicitly converted a major element of what makes subjectivity into mathematical form – specifically, we explicitly model the human concept vocabulary H – thus opening the door to understanding its consequences using powerful mathematical tools, especially tools from causal representation learning. The most immediate consequence is that we were able to define concept interpretability in terms of an intuitive name transfer process, as well as a formal property (alignment) that must hold if we wish the name transfer process to succeed. Specifically, our framework for human-interpretable representations includes two important aspects: (1) a formal account the purely objective properties of the representations (which is the core of the causal framework we proposed), and (2) the human element, itself split into two pieces (i) the human concept vocabulary, and (ii) the notion of simulatability (by itself not new, but used here to back up the name transfer process). We also discussed (in Section 5) how other human factors, besides alignment, can influence the success of the name transfer process. Still, we kept our formalization as simple as possible (and in fact as close to causal representation learning as possible) to facilitate analysis and follow-up work using well-known tools from representation learning. To the best of our knowledge, this is novel. We have briefly discussed this point and included the references you provided, in section 6.
- Communication via human-interpretable representations. Definition 5 is vague by design - as not all necessary notions have been introduced yet at this point. Our intention was to prime the reader’s intuition with this definition, and then explain the various terms along Section 3, piece by piece. It is true that this definition is about communication, while the follow-up text ends up discussing alignment. The idea is that, as long as the concept vocabularies of two communicating agents are aligned (meaning that, roughly speaking, the two agents assign the very same semantics to each of the concepts in the vocabulary), then any message expressed using these concepts will have the same meaning for both agents, thus enabling communication. Then, in order to assess whether the concepts sharing the same name also share the same semantics, we introduced a name transfer process that ensures that, regardless of what input the two agents want to talk about, the concept-level description they would assign this input is either the very same or there exists a simple correspondence between the descriptions given by the two agents. (Both in an observational setting in which inputs are drawn from the world and an interventional setting in which the human has control of what inputs the machine observes.) Then we have shown that, if alignment holds, this process can succeed (in idealized conditions). In summary, if alignment holds, the concepts in the vocabulary have the same meaning for both agents, and the agents can communicate successfully regardless of what input they have to talk about. We have made it clear in the text that the definition 3 is meant to be an intuitive statement.
- The challenge in accessing the user concepts. Good point. Let us start with a clarification: when we talk about the human concept vocabulary H, we refer to the set of those concepts that humans can verbalize and use for communication with each other - that is, chiefly nouns and adjectives. A part of our main message is that without any reference (that is what concepts are relevant and their structure) establishing the interpretability of the representations is not possible. This contrasts with the mainstream tendency of learning concepts in a fully unsupervised manner.
It is true that these annotations are often not available in machine learning tasks like classification or regress: all we have are inputs and labels. Yet, humans can be asked to provide annotations based on these concepts. This is not commonly done because it is expensive - yet, it is also necessary to measure alignment. The (very concrete) risk is that humans and machines end up exchanging messages based on concepts that are not aligned: when the machine talks about a “red ball”, it does not mean the same thing that a human means. This dramatically complicates trustworthiness.
As for how to collect concept annotations: 1) Approaches like GlanceNets, Concept-based Models, and more, require annotations, meaning that at least in some cases these annotations are available; 2) It is possible to augment CBMs and CBEs with an interactive concept elicitation mechanism whereby they ask for concept annotations whenever they are unsure they have fully grasped the meaning of a concept (e.g., when they receive an input that they cannot describe conceptually with a sufficient degree of confidence), akin to active learning. Recent works in the CBM literature have started tackling this problem, although more work needs to be done, and we plan to pursue this research direction in future work. We briefly discussed this at the very end of Section 5, adding a link to these new works.
Finally, notice that alignment can be readily estimated for concepts shared by two models (e.g., two LLMs, or an LLM and a smaller distilled model), thus making it possible to ascertain whether these models assign the same meaning to the concepts they encode. - Complexity and scalability of evaluating D1 and D2. We are aware that the complexity of evaluating disentanglement of the representation is an open problem that is central in the study of metrics, see [ref 67]. Depending on the metric adopted for estimating disentanglement, the sample set can vary (see REF sec. 6.3). The sample number for estimating EMPIDA – which is the metric more closely related to our construction – grows exponentially with the number of possible realizations, although it is claimed in the original main paper that by choosing a proper discretization of G and restricting to single factor interventions does not require an excessive number of data. We expect that evaluating D2 in eq. (19) would have the same sample complexity. Other approaches, like DCI, require fitting a model from M to G so that ultimately the number of total data would be proportional to the number of parameters of the model. In principle, using a normalizing flow (that is invertible by construction) or a linear model can be a way to evaluate both disentanglement and invertibility/monotonicity. In terms of scalability, EMPIDA requires O(N) computations, where N is the number of samples, whereas DCI is more costly. Similarly, we expect that evaluating D2 as in eq. (19) would require the same complexity, since it builds on the same interventions analysis of EMPIDA.
As we mentioned in the discussion on the necessity of alignment, it is also the case that not all combinations of concepts have to be explored and restricted to the support of the model training distribution. We plan to design computationally suitable metrics for (block-) alignment in the near future. We added this discussion in Section 5.2.
Round 2
Reviewer 1 Report
Comments and Suggestions for Authors
I think the paper should be accepted without any further comments.
Reviewer 3 Report
Comments and Suggestions for Authors
The authors have adequately addressed my comments. My only reservation concerns the claim made by the authors that their mathematical formulation captures (some elements of) human subjectivity, which I believe is a rather strong claim. Nonetheless, this is more of a philosophical point that does not detract from the technical merit of the article in and of itself. Therefore, I believe that the article could be accepted.